# Yeast eIF4A enhances recruitment of mRNAs regardless of their structural complexity

Paul Yourik[1], Colin Echeverría Aitken[1†], Fujun Zhou[1], Neha Gupta[1], Alan G Hinnebusch[2]*, Jon R Lorsch[1]*

[1]Laboratory on the Mechanism and Regulation of Protein Synthesis, Eunice Kennedy Shriver National Institute of Child Health and Human Development, National Institutes of Health, Bethesda, United States; [2]Laboratory of Gene Regulation and Development, Eunice Kennedy Shriver National Institute of Child Health and Human Development, National Institutes of Health, Bethesda, United States

**Abstract** eIF4A is a DEAD-box RNA-dependent ATPase thought to unwind RNA secondary structure in the 5'-untranslated regions (UTRs) of mRNAs to promote their recruitment to the eukaryotic translation pre-initiation complex (PIC). We show that eIF4A's ATPase activity is markedly stimulated in the presence of the PIC, independently of eIF4E•eIF4G, but dependent on subunits i and g of the heteromeric eIF3 complex. Surprisingly, eIF4A accelerated the rate of recruitment of all mRNAs tested, regardless of their degree of structural complexity. Structures in the 5'-UTR and 3' of the start codon synergistically inhibit mRNA recruitment in a manner relieved by eIF4A, indicating that the factor does not act solely to melt hairpins in 5'-UTRs. Our findings that eIF4A functionally interacts with the PIC and plays important roles beyond unwinding 5'-UTR structure is consistent with a recent proposal that eIF4A modulates the conformation of the 40S ribosomal subunit to promote mRNA recruitment.

DOI: https://doi.org/10.7554/eLife.31476.001

*For correspondence:
ahinnebusch@nih.gov (AGH);
jon.lorsch@nih.gov (JRL)

Present address: †Biology Department, Vassar College, New York, United States

## Introduction

The goal of translation initiation is to assemble the ribosomal initiation complex containing the methionyl initiator tRNA (Met-tRNA$_i$) at the translation start site on an mRNA. The process begins when the small (40S) subunit of the ribosome binds eIF1, eIF1A, eIF2, GTP, Met-tRNA$_i$, eIF3, and eIF5, to assemble the 43S PIC (*Dever et al., 2016*). eIF1 and eIF1A bind near the P and A sites of the 40S subunit, respectively, and promote loading of the ternary complex (TC) comprising eIF2, GTP, and Met-tRNA$_i$. eIF5 is the GTPase-activating protein (GAP) for eIF2 and stimulates GTP hydrolysis (*Das et al., 1997, 2001; Paulin et al., 2001*); however, irreversible release of the inorganic phosphate is inhibited at this stage of the pathway (*Algire et al., 2005*). The heteromultimeric factor eIF3 has multiple interactions with the PIC and is involved in nearly every step of translation initiation (*Aitken and Lorsch, 2012; Valásek, 2012*).

The 43S PIC is assembled in an 'open' conformation (*Fekete et al., 2007; Llácer et al., 2015; Maag et al., 2005, 2006; Passmore et al., 2007; Pestova et al., 1998a*) that can bind an mRNA in a process called mRNA recruitment. eIF3 and a set of mRNA recruitment factors – eIF4A, eIF4E, eIF4G, and eIF4B – facilitate this step (*Mitchell et al., 2011*). After initial mRNA loading, the PIC remains in an open conformation and scans the 5'- untranslated region (UTR) of the mRNA for the start codon, usually an AUG (*Hinnebusch, 2014*). Recognition of the start codon by the Met-tRNA$_i$ triggers a series of key steps – eviction of eIF1, movement of the C-terminal tail of eIF1A out of the P site, and subsequent release of the previously-hydrolyzed inorganic phosphate by eIF2 – ultimately

shifting the PIC from an 'open' to a 'closed' conformation (*Dever et al., 2016*; *Hussain et al., 2014*; *Llácer et al., 2015*). The 48S PIC thus formed is committed to the selected start codon and joining with the large (60S) subunit of the ribosome to form the final 80S initiation complex (*Acker et al., 2009*; *Dever et al., 2016*).

Whereas a combination of genetic, biochemical, and structural approaches have illuminated the molecular details of PIC formation and start-codon selection, the intermediate events of mRNA recruitment are less well understood (*Aitken and Lorsch, 2012*). We demonstrated that the individual absence of eIF4A, eIF4B, eIF3, or the eIF4G•eIF4E complex greatly reduces the extent, the rate or both of mRNA recruitment to the PIC in a *S. cerevisiae in vitro* reconstituted translation initiation system (*Mitchell et al., 2010*). Structural and biochemical work indicates that eIF3 binds on the solvent side of the 40S subunit but its five core subunits have multiple interactions with other components of the PIC including distinct interactions with the mRNA near the entry and exit channels of the ribosome (*Aitken et al., 2016*; *Llácer et al., 2015*). Also, yeast eIF4B binds directly to the 40S ribosomal subunit, modulates the conformation of the ribosome near the mRNA entry channel (*Walker et al., 2013*), and has a functional interaction with eIF4A (*Andreou and Klostermeier, 2014*; *Harms et al., 2014*; *Park et al., 2013*; *Walker et al., 2013*). In contrast, eIF4A has not been shown to bind stably to the PIC but forms a heterotrimeric complex with eIF4G and eIF4E (eIF4A•4G•4E), collectively referred to as eIF4F, which interacts with the mRNA.

mRNA can form stable secondary structures via local base-pairing of complementary sequences but also has a natural tendency to form global structure, marked by a combination of entangled or compacted conformations inherent to polymers longer than their persistence lengths (*Chen et al., 2012*) and the sum of many, possibly dynamic, individual base-pairing and other (tertiary) interactions between both neighboring and distant parts of the molecule (*Halder and Bhattacharyya, 2013*). Hairpin structures in the 5'-UTR are generally inhibitory to translation initiation and the prevailing model of mRNA recruitment suggests that eIF4A – localized to the 5'-end via the eIF4G–eIF4E–5'-m$^7$G-cap chain of interactions – unwinds these hairpins to allow PIC attachment (*Merrick, 2015*; *Pelletier and Sonenberg, 1985*; *Ray et al., 1985*; *Svitkin et al., 2001*; *Yoder-Hill et al., 1993*).

eIF4A is a DEAD-box RNA-dependent ATPase thought to act as an RNA helicase (*Andreou and Klostermeier, 2013*; *Linder et al., 1989*; *Schreier et al., 1977*). It cooperatively binds RNA and ATP with no apparent RNA sequence specificity, although, recent ribosome profiling experiments suggested that eIF4A may have a preference for binding polypurine sequences in vivo (*Iwasaki et al., 2016*). eIF4A is thought to disrupt structures by local strand separation, possibly by bending the RNA duplex (*Henn et al., 2012*; *Linder and Jankowsky, 2011*). Intermediate states of the catalytic cycle may have higher affinities for RNA; however, ATP hydrolysis to ADP causes a decrease in affinity of eIF4A for RNA, which may allow recycling of the factor (*Andreou and Klostermeier, 2013*; *Jankowsky, 2011*; *Liu et al., 2008*; *Lorsch and Herschlag, 1998a*; *Lorsch and Herschlag, 1998b*; *Pause et al., 1993*). Several studies demonstrated that both yeast and mammalian eIF4A are able to disrupt short RNA duplexes in vitro (*Rajagopal et al., 2012*; *Ray et al., 1985*; *Rogers et al., 1999*) and more recent work suggests that mammalian eIF4A can unwind a large hairpin when the RNA is stretched between two tethers (*García-García et al., 2015*). Also, sensitivity to inhibition of translation by a dominant negative mutant of mammalian eIF4A is correlated with the degree of secondary structure in the 5'-UTR, supporting a role for eIF4A in removing structures from the 5'-end of mRNAs (*Svitkin et al., 2001*).

Nonetheless, eIF4A is a slow helicase when unwinding stable duplexes in vitro (*García-García et al., 2015*; *Lorsch and Herschlag, 1998b*; *Rajagopal et al., 2012*; *Rogers et al., 1999*) and it is unclear whether it could support in vivo rates of translation initiation in the range of 10 min$^{-1}$ (*Palmiter, 1975*; *Shah et al., 2013*; *Siwiak and Zielenkiewicz, 2010*) if this is its function. Several more potent helicases appear to promote translation by resolving stable structural elements (*Parsyan et al., 2011*), raising the possibility that eIF4A instead performs a distinct role during initiation on all mRNAs (*Gao et al., 2016*). A study in yeast employing ribosome profiling to compare the in vivo effects of inactivating eIF4A versus Ded1 – a considerably more robust DEAD-box RNA helicase – demonstrated that translation of mRNAs with 5'-UTRs possessing high degrees of structure tend to be specifically dependent on Ded1, but relatively few mRNAs are similarly hyper-dependent on eIF4A, despite comparable requirements for the two helicases in maintaining global translation initiation. Consistent with this, analysis of reporter mRNAs harboring inhibitory hairpin structures

indicated that yeast eIF4A is either ineffective or dispensable in the presence of Ded1 in resolving stable, local secondary structures in the 5'-UTR (*Sen et al., 2015*). In a separate study, minor decreases in the normally high cellular levels of yeast eIF4A also resulted in depressed global rates of translation (*Firczuk et al., 2013*), providing additional evidence that eIF4A may be important for translation of all mRNAs and must be present in excess of other components of the translational apparatus. Consistent with the idea that it acts generally rather than by disrupting specific secondary structures in 5'-UTRs, eIF4A has been shown to promote translation of an mRNA possessing a short (8 nucleotide (nt)) 5'-UTR (*Blum et al., 1992*) and a viral mRNA with a low degree of structure in its 5'-UTR (*Altmann et al., 1990*), and to stimulate 48S PIC formation on mRNAs with little secondary structure in their 5'-UTRs (*Pestova and Kolupaeva, 2002*). Recent experiments using a fluorescence-based equilibrium binding assay indicated that eIF4A enhances the affinity of mammalian PICs for both a natural and an unstructured model mRNA, although ATP hydrolysis was only required for this effect with the natural mRNA (*Sokabe and Fraser, 2017*). How the helicase and ATPase activities of eIF4A, alone or as a part of eIF4F, contribute to its role in promoting initiation on diverse mRNAs remains unclear.

Here – using an in vitro translation initiation system reconstituted from purified *S. cerevisiae* components – we examined the interplay among eIF4A, the eIF4G•eIF4E complex (hereafter referred to as 'eIF4G•4E'), the mRNA, and the PIC in eIF4A-catalyzed ATP hydrolysis and mRNA recruitment. We monitored eIF4A ATPase in the context of the PIC and asked how eIF4A activity might be utilized for the recruitment of mRNAs possessing various degrees of structure, ranging from the natural (structured) *RPL41A* mRNA to short model mRNAs comprised mostly of CAA-repeats expected to lack significant structure (*Sobczak et al., 2010*; *Zuker, 2003*) beyond fluctuations in polymer conformation or transient interactions (*Chen et al., 2012*) (hereafter called 'unstructured' mRNA). We show eIF4A and eIF4A•4G•4E are both faster ATPases in the presence of the PIC and that this effect is independent of the stimulation provided by RNA or eIF4G•4E, but is dependent on the 3g and 3i subunits of eIF3. eIF4A increases the rate of recruitment for all mRNAs tested, ranging from the natural *RPL41A* mRNA to short unstructured messenger RNAs. Structures in the 5'-UTR and on the 3' side of the start codon synergistically inhibit mRNA recruitment in a manner relieved by eIF4A. Our data indicate that eIF4A can relieve inhibition of mRNA recruitment arising from structure created by elements throughout the length of the mRNA rather than only resolving secondary structures in the 5'-UTR. Overall, these results are consistent with a recent model suggesting that eIF4A may modulate the conformation of the 40S ribosomal subunit to promote mRNA recruitment (*Sokabe and Fraser, 2017*).

## Results

### ATP hydrolysis by eIF4A promotes recruitment of the natural mRNA *RPL41A* as well as a short, unstructured model mRNA

To better understand how eIF4A-catalyzed ATP hydrolysis is related to the removal of RNA structure and mRNA recruitment, we compared the kinetics of recruitment of the natural mRNA *RPL41A* (possessing structural complexity throughout its length; *Figure 1—figure supplement 1C*) with a 50 nucleotide (nt) model mRNA made up of CAA-repeats and an AUG codon at positions 24–26 (CAA 50-mer) (*Aitken et al., 2016*) (*Figure 1—figure supplement 1A–B*; see RNAs 1 and 10 in *Supplementary file 1*). As for most natural mRNAs, *RPL41A* is thought to have numerous base-pairing interactions throughout its length while the CAA 50-mer is expected to have little, if any, structure (*Sobczak et al., 2010*; *Zuker, 2003*) (*Figure 1—figure supplement 1*).

mRNA recruitment experiments were performed as described previously, using an in vitro-reconstituted *S. cerevisiae* translation initiation system and single turnover kinetics (*Mitchell et al., 2010*; *Walker et al., 2013*). Briefly, 43S PICs containing 40S, TC, eIF1, eIF1A, eIF5, and eIF3 were formed in the presence of saturating levels of eIFs 4A, 4B, 4E, and 4G (see mRNA Recruitment Assay in Materials and methods). eIF4A, 4G, and 4E are assumed to form the heterotrimeric eIF4F complex under experimental conditions (*Mitchell et al., 2010*). Reactions were initiated by simultaneous addition of ATP and an mRNA labeled with a [$^{32}$P]–7-methylguanosine (m$^7$G) cap, enabling mRNA recruitment to the PICs and formation of 48S complexes. Reaction timepoints were acquired by mixing an aliquot with a 25-fold excess of a non-radioactive ('cold') capped mRNA identical to the

labeled one in the reaction, effectively stopping further recruitment of radiolabeled mRNA. The rate of dissociation of recruited mRNAs from the PIC in the presence of the cold chase mRNA was negligible for all mRNAs in the study (data not shown). Free mRNA and 48S complexes were resolved via gel electrophoresis on a native 4% THEM polyacrylamide gel (*Acker et al., 2007*; *Mitchell et al., 2010*).

We first compared the kinetics of recruitment for *RPL41A* with CAA 50-mer in the presence and absence of ATP (*Figure 1*). In the presence of saturating ATP, the rate of recruitment of *RPL41A* was $0.74 \pm 0.01$ min$^{-1}$ with an endpoint in excess of 90%. In contrast, in the presence of ADP, and in reactions lacking either nucleotide or eIF4A, less than 20% of *RPL41A* mRNA was recruited after 6 hr, indicating a dramatically lower rate that could not be measured accurately due to the low reaction endpoint (*Figure 1A,C*). The CAA 50-mer was recruited in the absence of eIF4A and ATP at rates of about 0.90 min$^{-1}$, likely due to lack of significant structure, reaching endpoints around 80%. Surprisingly, the addition of eIF4A and ATP stimulated recruitment of the CAA 50-mer to a rate that could not be measured accurately by manually quenching the reaction; however, we estimate that the increase in rate was at least 4-fold ($4.0 \pm 0.6$ min$^{-1}$) and yielded an endpoint of 90% (*Figure 1B, C*).

To determine whether ATP hydrolysis is required for the stimulation of the rate of mRNA recruitment that we observed, we next measured the rate of recruitment with the non-hydrolyzable ATP analogs ADPCP and ADPNP, as well as with the slowly-hydrolyzable analog ATP-γ-S (*Peck and Herschlag, 2003*). Neither ADPCP nor ADPNP supported stimulation of the recruitment of either mRNA by eIF4A, producing rates that were comparable to the observed rates measured in the absence of nucleotide or eIF4A (*Figure 1*; compare grey and blue curves to red and purple). In the presence of ATP-γ-S, recruitment of *RPL41A* and CAA 50-mer was 39-fold ($0.019 \pm 0.001$ min$^{-1}$) and nearly 2-fold ($2.3 \pm 0.2$ min$^{-1}$) slower, respectively, than in the presence of ATP; however, both mRNAs achieved endpoints of approximately 80%, consistent with previous observations that eIF4A is capable of utilizing ATP-γ-S (*Peck and Herschlag, 2003*). Taken together, these results suggest that ATP hydrolysis by eIF4A stimulates the recruitment of both a natural mRNA harboring structure throughout its sequence and the short unstructured CAA 50-mer.

## The ATPase activity of eIF4A•4G•4E is increased by the PIC

Because of the importance of ATP hydrolysis for the ability of eIF4A to stimulate recruitment of both structured and unstructured mRNAs, we next investigated how the mRNA and PIC influence the ATPase activity of eIF4A and the eIF4A•4G•4E heterotrimer. Single turnover conditions in which the concentration of enzyme is saturating and greater than the concentration of the substrate, similar to those employed in the mRNA recruitment experiments above (*Figure 1*), would be ideal to study the ATPase activity for comparison. However, such an approach was technically not feasible because the low affinity of yeast eIF4A for ATP (*Rajagopal et al., 2012*) made it impossible to create conditions where eIF4A is saturating for ATP binding and stoichiometric to or in excess of ATP. Using saturating ATP under experimentally accessible concentrations of PIC and eIF4A led to a situation in which the first few turnovers of ATP hydrolysis produced by any possible eIF4A•PIC complexes would be below the limit of detection using either radioisotope or spectrophotometric ATPase assays. In addition, based on the high (~5 µM) concentration of eIF4A required to achieve maximal rates of mRNA recruitment, we also could not achieve a situation in which PICs were saturating over eIF4A because such concentrations of PICs are not experimentally achievable. Thus, we were not able to perform pre-steady state kinetic ATPase experiments. Instead, to inquire whether the ATPase activity of eIF4A is affected by the presence of the PIC, we used multiple turnover conditions in which [eIF4A] << [ATP] in the in vitro reconstituted translation initiation system. Although steady-state kinetics do not allow direct comparison to single-turnover mRNA recruitment assays, the approach still enables detection of the effects of other components of the system on repeated cycles of ATP hydrolysis by eIF4A and eIF4A•4G•4E. Thus, if the PIC in the absence of mRNA (i.e., prior to mRNA recruitment) or the presence of mRNA (i.e., during and after mRNA recruitment) promotes a state of eIF4A with altered ATPase activity, it should be possible to detect it using this approach.

ATPase was monitored with an enzyme-coupled assay in which pyruvate kinase and lactate dehydrogenase are used to regenerate ATP from ADP and, in the process, oxidize NADH to NAD$^+$, producing a change in absorbance at 340 nm (*Bradley and De La Cruz, 2012*; *Kiianitsa et al., 2003*); (*Figure 2—figure supplement 1A*; see Materials and methods). eIF4A is an RNA-dependent

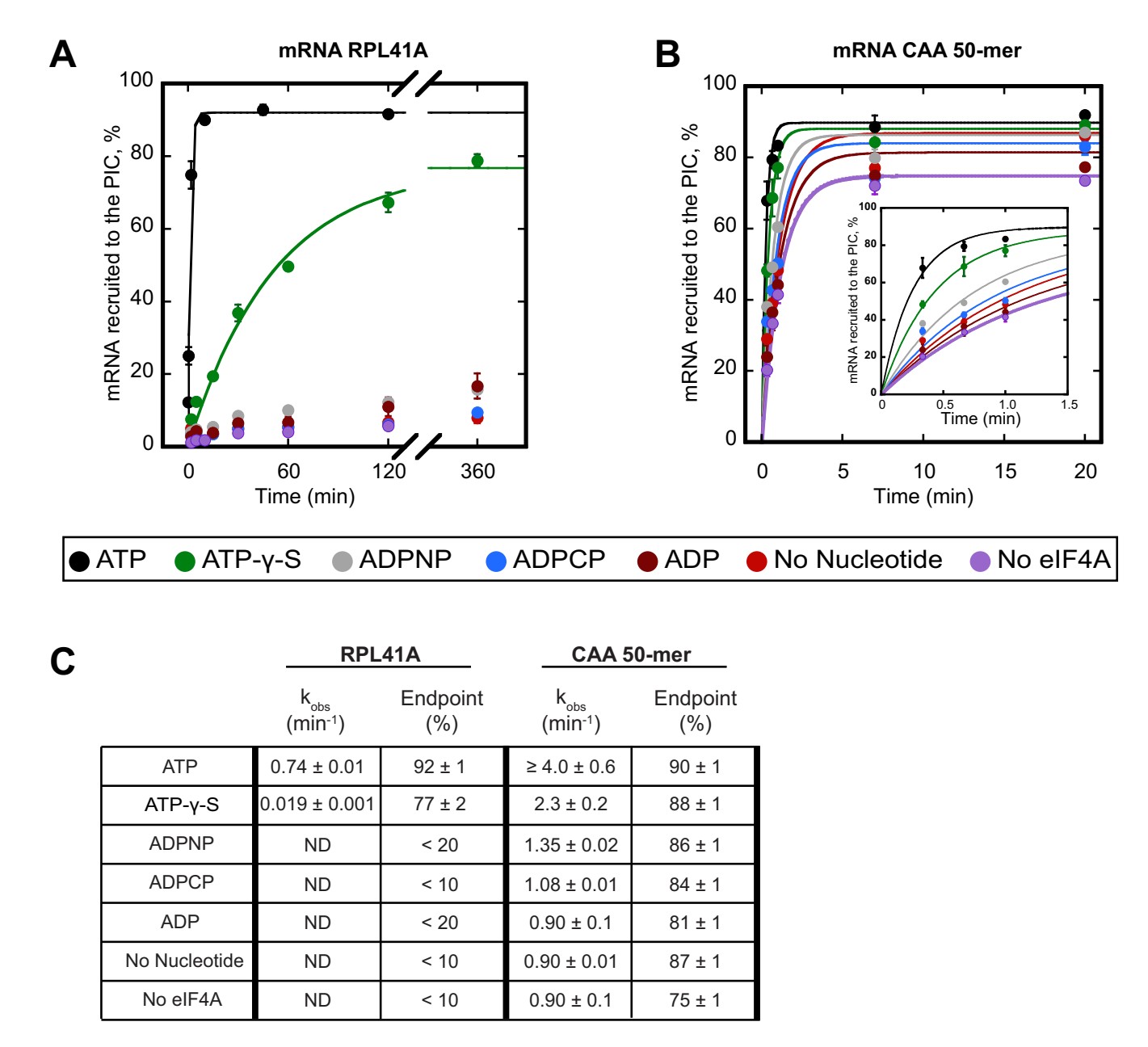

**Figure 1.** ATP hydrolysis by eIF4A stimulates recruitment of natural mRNA *RPL41A* as well as a synthetic 50-mer made up largely of CAA-repeats, presumed to be unstructured, with an AUG start codon 23 nucleotides from the 5'-end (CAA 50-mer). The concentration of ATP and analogs was 2 mM. (**A**) Percentage of *RPL41A* recruited to the PIC versus time. (**B**) Percentage of CAA 50-mer recruited to the PIC versus time. The larger plot shows the time-course up to 20 min. The inset shows the first 1.5 min for clarity. With the CAA 50-mer in the presence of ATP and eIF4A, reaction rates are at the limit of our ability to measure by hand but the results provide an estimate of the value. (**C**) Observed rate constants ($k_{obs}$) and reaction endpoints from the data in panels A and B. All data in the figure are mean values (n ≥ 2) and error bars represent average deviation of the mean. With *RPL41A*, rates with ADPNP, ADPCP, ADP, no nucleotide, and no eIF4A could not be measured accurately due to low endpoints ('ND').

DOI: https://doi.org/10.7554/eLife.31476.002

The following source data and figure supplement are available for figure 1:

**Source data 1.** Individual measurements of the percent of *RPL41A* and CAA 50-mer recruited to the PIC over time.
DOI: https://doi.org/10.7554/eLife.31476.004

**Figure supplement 1.** RPL41A can form structural elements throughout its length whereas the CAA 50-mer has a low propensity to form structures.
DOI: https://doi.org/10.7554/eLife.31476.003

ATPase, thus the presence of mRNA stimulates the rate of ATP hydrolysis. Reactions with varying concentrations of mRNA were assembled in a 384-well plate and initiated by addition of saturating ATP (5 mM). NADH absorbance at 340 nm was recorded every 20 s using a microplate reader. By titrating mRNA, we determined the first-order rate constant ($k_{cat}$) of ATP hydrolysis at saturating mRNA and ATP concentration, as well as the concentration of mRNA needed to achieve the half-maximal velocity of ATP hydrolysis ($K_m^{RNA}$). For eIF4A alone (5 µM) the $k_{cat}$ was 0.48 ± 0.04 min$^{-1}$ (*Figure 2A*), comparable to a previously reported value of 0.20 ± 0.05 min$^{-1}$ (*Rajagopal et al., 2012*). Also, congruent with previous findings (*Hilbert et al., 2011*; *Oberer et al., 2005*; *Rajagopal et al., 2012*), the addition of co-purified full length eIF4G1 and eIF4E (eIF4G•4E) to eIF4A, forming the eIF4A•4G•4E heterotrimer, resulted in a 6.5-fold increase in the $k_{cat}$ (*Figure 2A*, open squares vs. *Figure 2B*, closed squares). Whereas addition of saturating mRNA increased the ATPase activity of eIF4A alone by only 2-fold in the absence of eIF4G•4E (*Figure 2A*), a larger (15-fold) increase in $k_{cat}$ occurred in the presence of eIF4G•4E (*Figure 2B*, closed squares).

Addition of the PIC to eIF4A•4G•4E in the absence of mRNA increased the $k_{cat}$ 24-fold over the value in the absence of the PIC (*Figure 2B*; compare closed circles and closed squares, respectively, at 0 µM mRNA). This enhancement could be due to the presence of rRNA and tRNA in the PIC components. However, at saturating concentrations of mRNA, the PIC components still enhanced the $k_{cat}$ for ATP hydrolysis by eIF4A•4G•4E by 3.4-fold (*Figure 2B*, compare closed circles to closed squares), indicating that there is an enhancement of ATPase activity not due to non-specific stimulation by RNA but instead caused by one or more components of the PIC. Leaving out 40S subunits from the PIC components resulted in a 2-fold lower $k_{cat}$ at saturating mRNA compared to the value observed in the presence of a Complete PIC (*Figure 2B*; compare closed and open circles). The fact that this value is less than the full 3.4-fold effect suggests that other components of the system can provide some modest activation of the ATPase of eIF4A•4G•4E in the absence of the Complete PIC. As described below, it is noteworthy that the PIC confers an ~6 fold increase in $k_{cat}$ for eIF4A in the absence of eIF4G•4E, which is comparable to the ~4 fold increase conferred by eIF4G•4E itself (*Figure 2C,D* and *Figure 2—figure supplement 1C,D*; compare '4A' to 'Complete PIC – eIF4G•4E' and '4A' to '4A•4G•4E'). Replacing capped *RPL41A* mRNA with uncapped mRNA yielded similar results (*Figure 2—figure supplement 1B*), indicating that the 5'-cap is not critical for the observed stimulation by the PIC or mRNA.

In order to determine which components are responsible for this stimulation of the ATPase activity of eIF4A•4G•4E, we repeated the experiments at a saturating mRNA concentration (0.5 µM), omitting various components alone or in combination. We also varied the ATP concentration to determine both the $k_{cat}$ (*Figure 2C*) and the concentration of ATP required to achieve the half-maximal velocity ($K_m^{ATP}$) (*Figure 2—figure supplement 1*). The $k_{cat}$ for eIF4A•4G•4E was 2.4 ± 0.1 min$^{-1}$ and adding the 40S subunit and TC (comprising eIF2, GDPNP, and Met-tRNA$_i$) – which alone are not sufficient to form the PIC – did not have any additional effect (*Figure 2C*, compare,'4A•4G•4E' and 'TC, 40S, 4A•4G•4E'). However, as already noted in *Figure 2B*, adding the Complete PIC increased the $k_{cat}$ by 3.4-fold to 8.2 ± 0.3 min$^{-1}$ (*Figure 2C* and *Figure 2—figure supplement 1C*, compare,'4A•4G•4E' and 'Complete PIC'). Omission of eIF4A from an otherwise Complete PIC resulted in a ≥ 67 fold decrease in the rate of ATPase as compared to the Complete PIC (to the limit of detection of the assay), ruling out any significant ATPase contamination in any of the PIC components (*Figure 2C*, '−4A').

As described above, leaving out 40S subunits from the PIC components resulted in a 2-fold lower $k_{cat}$ at saturating mRNA compared to the value observed in the presence of a Complete PIC (*Figure 2B*; compare closed and open circles; and *Figure 2C*, 'Complete PIC' vs. '-40S'). This reduction is slightly less than the full 3.4-fold effect of the Complete PIC on the $k_{cat}$ of eIF4A•4G•4E (*Figure 2—figure supplement 1C*). Similarly, leaving out eIF2 decreased the $k_{cat}$ by ~2 fold (*Figure 2C*; compare solid red 'Complete PIC' to cross-hatched, light red '−40S' and '−2' bars), indicating that each of these core components of the PIC is needed for full stimulation, but suggesting that other components of the system can still provide some modest stimulation on their own. Leaving out eIF2 and the 40S subunit together had a similar effect. Omitting eIF3, which binds tightly to the 43S PIC (composed of eIFs 1, 1A, 3, 5, eIF2·GDPNP·Met-tRNA$_i$, and the 40S subunit,) and contacts both the 40S subunit and multiple factors, decreased ATPase stimulation by 2.5-fold (*Figure 2C*, compare solid red 'Complete PIC' bars to cross-hatched, light red '−3' bar). The combined absences of 40S subunits and eIF3 resulted in a 2.7-fold decrease in $k_{cat}$ as compared to the Complete PIC

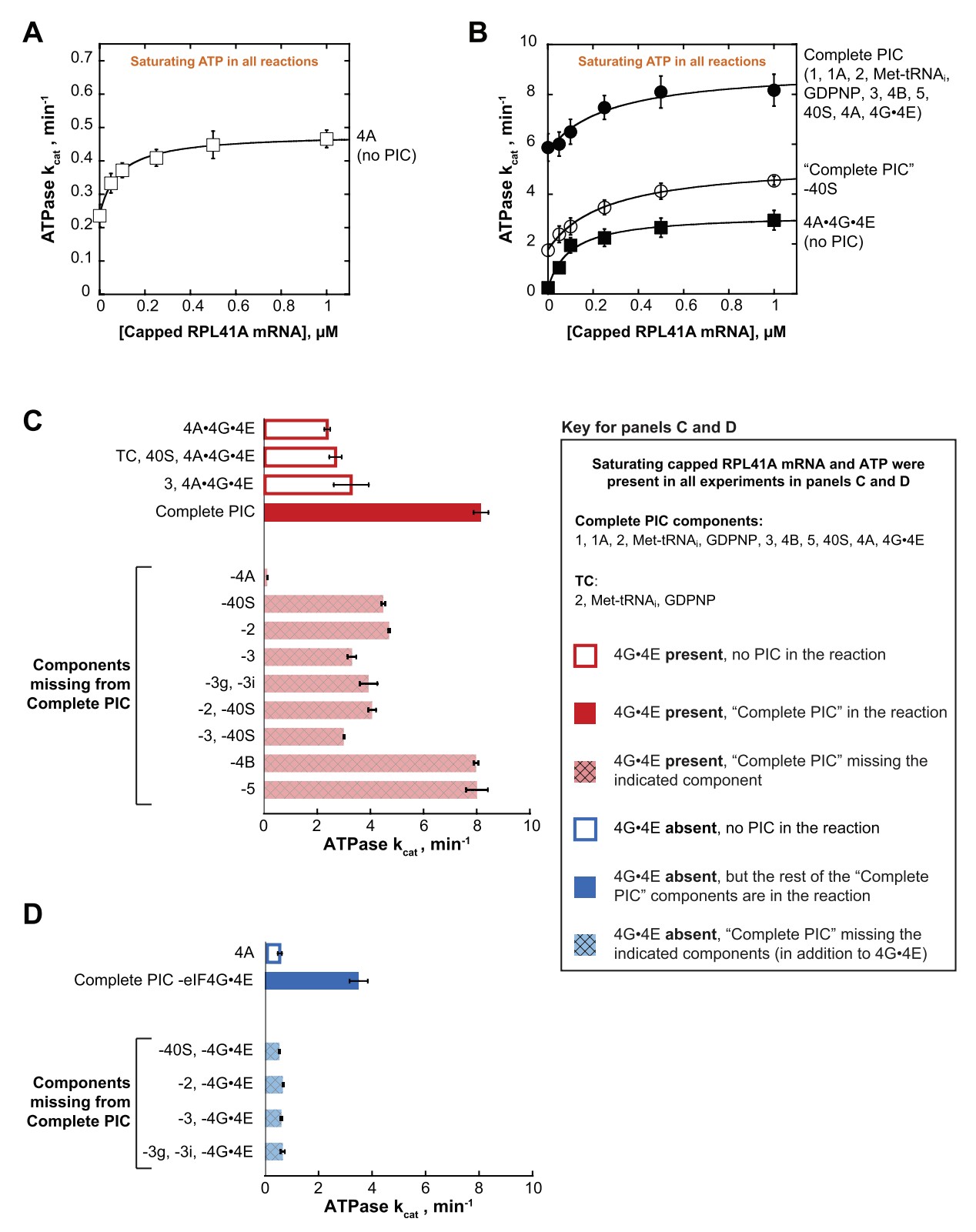

**Figure 2.** eIF4A and eIF4A•4G•4E ATPase activities are stimulated by the PIC. ATPase activity of 5 µM eIF4A in the presence of saturating (5 mM) ATP•Mg$^{2+}$ as a function of the concentration of capped *RPL41A* mRNA for (**A**) eIF4A (no PIC): $k_{cat}$ = 0.48 ± 0.04 min$^{-1}$, $K_m^{RNA}$ = 80 nM; (**B**) Solid black circles: Complete PIC (contains 5 µM eIF4A, 0.5 µM eIF4G•4E, 0.5 µM eIF4B, 0.5 µM eIF2, 0.5 µM Met-tRNA$_i$, 1 mM GDPNP•Mg$^{2+}$, 0.5 µM eIF3, 0.5 µM eIF5, 1 µM eIF1, 1 µM eIF1A, and 0.5 µM 40S subunits): $k_{cat}$ = 9.0 ± 0.8 min$^{-1}$, $K_m^{RNA}$ = 260 ± 40 nM. White circles: 'Complete PIC-40S' (contains all

*Figure 2 continued on next page*

Figure 2 continued

'Complete PIC' components except the 40S subunit): $k_{cat}$ = 5.3 ± 0.1 min$^{-1}$, $K_m^{RNA}$ = 270 ± 70 nM. Solid black squares: 4A•4G•4E alone (no PIC) (contains 5 µM eIF4A and 0.5 µM eIF4G•4E only): $k_{cat}$ = 3.2 ± 0.2 min$^{-1}$, $K_m^{RNA}$ = 100 ± 10 nM. (C) $k_{cat}$ values for ATP hydrolysis from the experiments described in (A) and (B) plus additional drop-out experiments. In all cases, saturating (0.5 µM) capped *RPL41A* mRNA was present. The components present in the reactions, as indicated to the left of each bar, were at the same concentrations as in (B). (D) ATPase activity of 5 µM eIF4A alone or in the presence of a PIC missing eIF4G•4E as well as other components, as indicated to the left of each bar, all measured in the presence of saturating (0.5 µM) capped *RPL41A* mRNA. The concentrations of all components, when present, were the same as in (B). Note that the axis scales in panels (C) and (D) are the same for ease of comparison. All data presented in the figure are mean values (n ≥ 2) and error bars represent average deviation of the mean.
DOI: https://doi.org/10.7554/eLife.31476.005

The following source data and figure supplement are available for figure 2:

**Source data 1.** Individual $k_{cat}$ measurements of ATP hydrolysis.
DOI: https://doi.org/10.7554/eLife.31476.007
**Source data 2.** Related to *Figure 2—figure supplement 1*.
DOI: https://doi.org/10.7554/eLife.31476.008
**Figure supplement 1.** Control and data tables for the ATPase experiments.
DOI: https://doi.org/10.7554/eLife.31476.006

(*Figure 2C*; '−2,–40S', '−3,–40S'), close to the full 3.4-fold activation, suggesting that eIF3 bound to a 43S PIC is the functional unit of activation. Consistent with this conclusion, eIF3 by itself was not sufficient to significantly increase the $k_{cat}$ of eIF4A•4G•4E (*Figure 2C*, compare 'eIF4A•4G•4E', '3, 4A•4G•4E' and 'Complete PIC').

In contrast to the core 43S PIC components and eIF3, leaving out eIF4B or eIF5 had no effect on the stimulation of the ATPase activity of eIF4A•4G•4E. Neither factor is required for formation of a stable PIC or binding of eIF3 to the complex, consistent with the proposal that it is the 43S PIC bound to eIF3 that stimulates the ATPase activity.

## eIF3 subunits g and i, critical for mRNA recruitment, are necessary for ATPase stimulation

*S. cerevisiae* eIF3 is comprised of 5 core subunits and is involved in numerous steps of translation initiation, including mRNA recruitment. Previous studies showed that yeast eIF3 is essential for mRNA recruitment both in vitro (*Mitchell et al., 2010*) and in vivo (*Jivotovskaya et al., 2006*), and that it stabilizes TC binding and promotes PIC interactions with the mRNA at both the mRNA entry and exit channels of the 40S subunit (*Aitken et al., 2016*). In particular, the eIF3 subunits 3g and 3i have been implicated in scanning and AUG recognition in vivo (*Cuchalová et al., 2010*) and are required for recruitment of *RPL41A* mRNA in vitro (*Aitken et al., 2016*; *Valásek, 2012*). Both subunits are thought to be located near the path of the mRNA on the ribosome, at either the solvent or intersubunit face of the 40S subunit, and may undergo large alterations in position during initiation (*Aylett et al., 2015*; *des Georges et al., 2015*; *Llácer et al., 2015*). In the presence of PICs formed with the heterotrimer of eIF3 subunits 3a, 3b, and 3c – but lacking the 3g and 3i subunits – the $k_{cat}$ was 3.9 ± 0.3 min$^{-1}$, which is similar to the rate constant observed in the absence of the entire eIF3 complex (*Figure 2C* and *Figure 2—figure supplement 1C*, '−3' vs. '−3g, −3i'). Thus, although the heterotrimeric eIF3a•3b•3 c subcomplex binds to the PIC under our experimental conditions (*Aitken et al., 2016*) it does not increase the ATPase $k_{cat}$, indicating an important role for the eIF3i and eIF3g subunits in promoting ATPase activity. One possible scenario is that eIF3g and eIF3i interact with eIF4A near the mRNA entry channel of the 40S subunit, either directly or indirectly, and this interaction promotes ATP hydrolysis and mRNA recruitment.

## The presence of the 43S PIC and eIF3 increases eIF4A ATPase activity in the absence of eIF4G•4E

We next asked if the PIC could stimulate the ATPase activity of eIF4A in the absence of eIF4G•4E. eIF4A on its own, in the presence of saturating *RPL41A* mRNA but in the absence of any PIC components or eIF4G•4E, had a $k_{cat}$ of 0.58 ± 0.08 min$^{-1}$ in these experiments (titrating the concentration of ATP). Addition of the PIC, without eIF4G•4E, resulted in a 6-fold increase in $k_{cat}$ over eIF4A alone (*Figure 2D* and *Figure 2—figure supplement 1D*; '4A' vs. 'Complete PIC −4G• 4E'). Remarkably, this stimulation is even greater than the stimulation of eIF4A•4G•4E by the PIC (3.4-fold) and of

eIF4A by the eIF4G•4E complex alone (4.0-fold; *Figure 2—figure supplement 1C,D*). In fact, in the presence of the PIC, eIF4G•4E only increases the $k_{cat}$ for eIF4A by 2.3-fold (*Figure 2C,D* and *Figure 2—figure supplement 1C,D*; compare 'Complete PIC' and 'Complete PIC – eIF4G•4E'), ~2 fold less than the effect of eIF4G•4E on eIF4A in isolation. Thus, the PIC components markedly enhance the activity of eIF4A even when it is not associated with the eIF4G•4E complex.

We next asked which PIC components are critical for this eIF4G•4E-independent stimulatory mechanism. Eliminating the 40S subunit, eIF2, eIF3, or the 3g and 3i subunits of eIF3 in the absence of eIF4G•4E abrogated all stimulation of ATPase activity, yielding $k_{cat}$ values the same as those observed with eIF4A alone (~0.6 min$^{-1}$; *Figure 2D*, compare dark blue bar to light blue cross-hatched bars). These data suggest that stimulation of the ATPase activity of eIF4A, in the absence of eIF4G•4E, requires the complete 43S PIC and eIF3, in particular subunits 3i and 3g. This result suggests the possibility that eIF4A can interact directly with one or more components of the PIC – for example, eIF3i and 3g – independently of interactions that might be mediated by eIF4G; and its interactions with the PIC or eIF4G•4E confer comparable stimulation of eIF4A's ATPase activity (6.0-fold vs. 4.0-fold).

## $K_m$ values suggest distinct mechanisms of eIF4A activation by the PIC components and eIF4G•4E

Formation of the eIF4A•4G•4E complex reduces the $K_m$ for ATP ($K_m^{ATP}$) in ATP hydrolysis by an order of magnitude, from 2500 µM to 250 µM (*Figure 2—figure supplement 1C,D*, compare '4A' to '4A•4G•4E'). In contrast to the effect of eIF4G•4E, adding the Complete PIC to eIF4A does not reduce the $K_m$ for ATP; and, as might be expected, leaving out the 40S subunits, eIF2 or eIF3 from the PIC does not increase the $K_m$ (*Figure 2—figure supplement 1C,D*). Thus, although both the PIC components and eIF4G•4E increase eIF4A's $k_{cat}$ for ATP hydrolysis, only eIF4G•4E reduces the $K_m$ for ATP, suggesting different mechanisms of stimulation. The effect of eIF4G•4E is consistent with the proposal that eIF4G acts as a 'soft clamp' to juxtapose the two eIF4A RecA-like domains to enhance ATP binding and catalysis (*Hilbert et al., 2011*; *Oberer et al., 2005*; *Schütz et al., 2008*). By contrast, the PIC components apparently act in a manner that enhances the rate-limiting step of ATP hydrolysis.

The $K_m^{ATP}$ values we measured for eIF4A alone are ~5 fold higher than those reported in the literature for yeast (*Blum et al., 1992*) and mammalian (*Lorsch and Herschlag, 1998b*; *Pause et al., 1993*; *Rogers et al., 1999*) eIF4A. Likewise, the $K_m^{ATP}$ value for eIF4A•4G•4E on its own was 2–3-fold higher than previously measured $K_m^{ATP}$ values for yeast eIF4A•4G•4E (*Rajagopal et al., 2012*). Although we do not know the reason for these discrepancies, it is possible that they are due to differences in buffer conditions (*Lorsch and Herschlag, 1998b*) or to differences in the RNA used (e.g., poly(U) or poly(A) vs. a structured mRNAs; [*Harms et al., 2014*]). In addition, there are some known differences between the behavior of yeast and mammalian eIF4A, including tighter binding between eIF4A and eIF4G•4E in mammals, that could result in species-specific differences in kinetic parameters (*Merrick, 2015*).

## eIF4A relieves inhibition of recruitment produced by structures throughout the length of mRNAs

Having established that ATP hydrolysis by eIF4A accelerates the rate of recruitment of both the natural mRNA *RPL41A* and an unstructured 50-mer made up of CAA-repeats (*Figure 1*) and that intact PICs activate steady-state ATP hydrolysis by eIF4A (*Figure 2*), we next set out to probe the effects of mRNA structure on the recruitment process and action of eIF4A. To this end, we created a library of in vitro transcribed and individually purified mRNAs spanning a range of structures and lengths. This library contains model mRNAs comprised almost entirely of CAA-repeats, containing or lacking a 9 base pair (bp) hairpin in the 5'-UTR, and/or a natural *RPL41A* mRNA sequence downstream of the AUG in place of CAA-repeats (*Figure 3A*; *Supplementary file 1*). The *RPL41A* sequence is expected to have structure throughout its length (*Figure 1—figure supplement 1C*). We measured the recruitment kinetics for each mRNA in the absence of eIF4A and as a function of eIF4A concentration (see 'mRNA Recruitment Assay' in Materials and methods) and determined the maximal rate ($k_{max}$) and concentration of eIF4A required to achieve the half-maximal rate ($K_{1/2}^{eIF4A}$) (*Figure 3—figure supplement 1*).

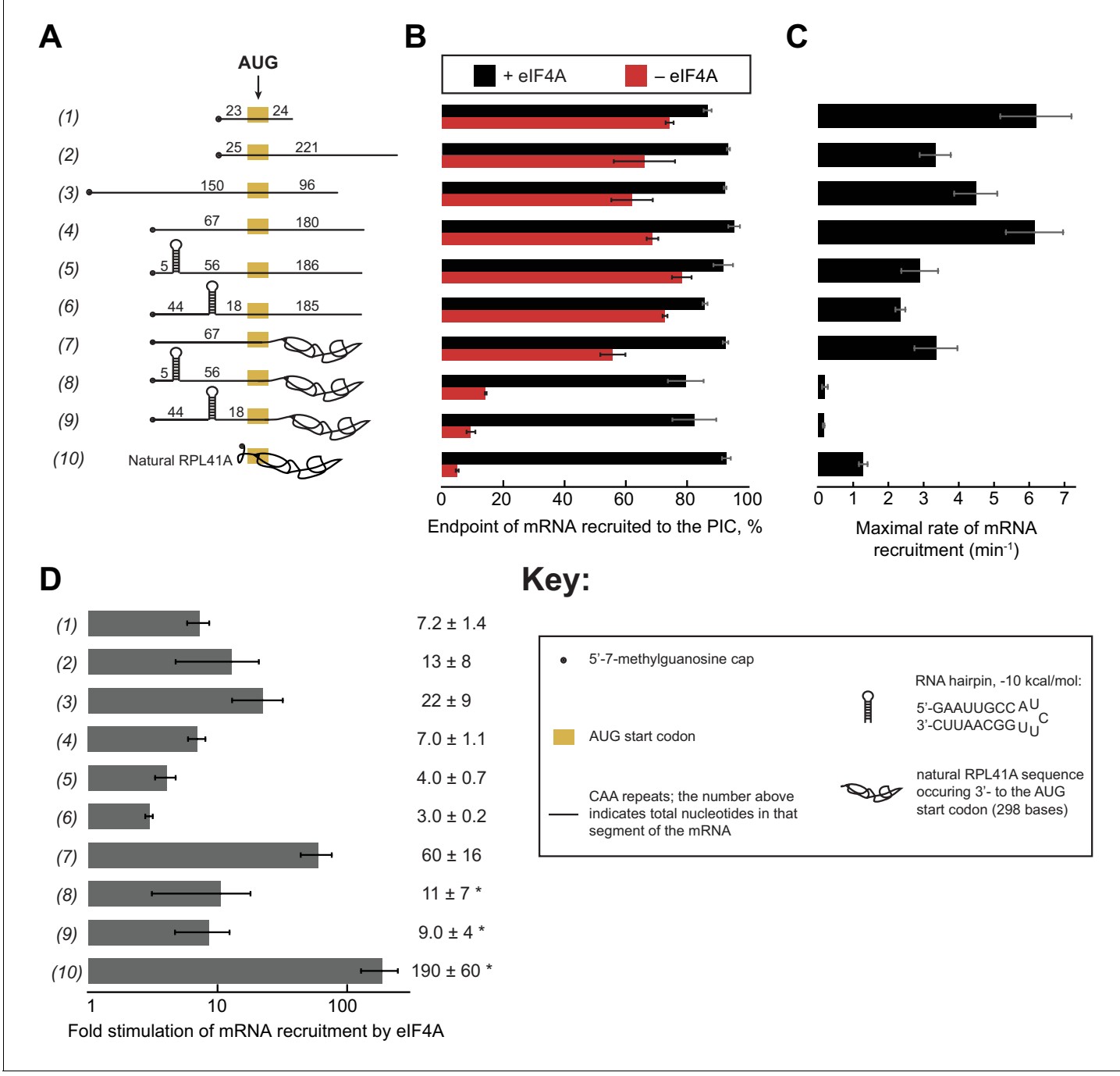

**Figure 3.** eIF4A stimulates recruitment of mRNAs regardless of their degree of structure. (A) Schematic of mRNAs used in the study. mRNAs were capped unless otherwise noted but do not contain a poly(A) tail. Numbers on the mRNA indicate the total number of nucleotides in the corresponding segment of the RNA. (B) Endpoints of recruitment to the PIC for mRNAs in (A) in the presence (black) or absence (red) of saturating eIF4A. mRNAs are listed in the same order as in (A). (C) Maximal rates of mRNA recruitment, $k_{max}$ (min$^{-1}$), measured for the mRNAs in (A) (the corresponding plots are shown in *Figure 3—figure supplement 1B–D*) listed in the same order as in (A). (D) eIF4A-dependent stimulation of mRNA recruitment: the maximal rate of mRNA recruitment at saturating eIF4A concentration divided by the observed rate in the absence of eIF4A (calculated from data in *Figure 3—figure supplement 1E*). Numbers to the left correspond to the mRNAs in (A). The asterisk (*) indicates that due to low recruitment endpoints in the absence of eIF4A, data for mRNAs 8–10 could not be fit with an exponential rate equation and thus the fold stimulation by eIF4A was estimated from comparison of initial rates in the presence of saturating eIF4A versus the absence of eIF4A. All data presented in the figure are mean values (n ≥ 2) and error bars represent average deviation of the mean.

DOI: https://doi.org/10.7554/eLife.31476.009

*Figure 3 continued on next page*

*Figure 3 continued*

The following source data and figure supplements are available for figure 3:

**Source data 1.** Individual measurements of endpoints and rates of mRNA recruitmentment for mRNAs 1–10.
DOI: https://doi.org/10.7554/eLife.31476.013
**Source data 2.** Related to *Figure 3—figure supplement 1*.
DOI: https://doi.org/10.7554/eLife.31476.014
**Figure supplement 1.** eIF4A promotes recruitment of both structured and CAA-repeats mRNAs.
DOI: https://doi.org/10.7554/eLife.31476.010
**Figure supplement 2.** Evidence that the designed hairpins in the 5'-UTRs of mRNAs 5 and 6 are formed and that mRNA 4 lacks secondary structure.
DOI: https://doi.org/10.7554/eLife.31476.011
**Figure supplement 3.** A change in the rate-limiting step for mRNA recruitment may be responsible for the effect of mRNA structure on the $K_{1/2}^{eIF4A}$ values.
DOI: https://doi.org/10.7554/eLife.31476.012

Using an eIF4A concentration determined to be saturating for all ten mRNAs (*Figure 3—figure supplement 1*), we observed recruitment endpoints between 85–95% with all mRNAs tested (*Figure 3B*, black bars). In the absence of eIF4A, however, the extent of recruitment varied widely among the mRNAs. Less than 10% of *RPL41A* mRNA was recruited in reactions lacking eIF4A (*Figure 3B*, RNA 10, red bar), consistent with the low levels of *RPL41A* recruitment we observed in the absence of ATP (*Figure 1A*). Varying the concentration of eIF4A yielded a $k_{max}$ of 1.3 ± 0.1 $min^{-1}$ (*Figure 3C*) and $K_{1/2}^{eIF4A}$ of 3.7 ± 1.0 μM for recruitment of *RPL41A* mRNA (*Figure 3—figure supplement 1B,E*). In the absence of eIF4A, time courses with *RPL41A* mRNA could not be accurately fit with a single-exponential kinetic model due to low reaction endpoints. However, comparison of estimated initial rates (no eIF4A, 0.22 ± 0.07 $min^{-1}$; saturating eIF4A, 41 ± 1 $min^{-1}$) of recruitment of *RPL41A* revealed that recruitment proceeds two orders of magnitude more rapidly in the presence of saturating levels of eIF4A versus in the absence of eIF4A (*Figure 3D*, RNA 10).

Consistent with our observation that the unstructured CAA 50-mer mRNA is efficiently recruited even in the absence of ATP, we observed 74 ± 1% recruitment of this mRNA in the absence of eIF4A (*Figure 3A–B*, RNA 1, red bar). Nonetheless, the addition of saturating eIF4A increased the extent of recruitment to 87 ± 1% (*Figure 3A–B*, RNA 1, black bar) consistent with our observation that the addition of ATP and eIF4A slightly elevates the extent of recruitment of this mRNA above the levels observed in the absence of ATP or eIF4A (*Figure 1B*). Beyond this modest stimulation of recruitment extent, saturating eIF4A markedly accelerated the rate of CAA 50-mer recruitment, yielding a $k_{max}$ of 6.2 ± 1.0 $min^{-1}$ (*Figure 3C*, RNA 1). The fold stimulation of the rate of mRNA recruitment by saturating eIF4A was calculated by dividing the $k_{max}$ by the observed rate of recruitment in the absence of eIF4A ($k_{obs}^{no\ eIF4A}$) (*Figure 3—figure supplement 1E*), yielding an ~7 fold increase in the rate of CAA 50-mer recruitment by eIF4A (*Figure 3D*, RNA 1). Relative to *RPL41A* mRNA, this acceleration is achieved at lower levels of eIF4A ($K_{1/2}^{eIF4A}$ of 0.70 ± 0.2 μM vs. 3.7 ± 1.0 μM with *RPL41A*; *Figure 3—figure supplement 1B,E*, RNAs 1 and 10). Because $K_{1/2}$ is a kinetic constant and may reflect changes in rate-limiting steps from mRNA recruitment to a later step or combination of steps as the concentration of eIF4A is increased, the affinity of eIF4A for eIF4G•4E or for the PIC cannot be inferred from the $K_{1/2}^{eIF4A}$; however, the constant does give insight into the dependence of the rate of the reaction on the concentration of eIF4A, and how varying features of the system such as the degree of structure in the mRNA affect this dependence.

To compare the CAA 50-mer with a longer mRNA, we increased the total mRNA length to 250 nucleotides (250-mer) by adding 200 nucleotides of CAA-repeats downstream of the AUG (*Figure 3A*, RNA 2). In the absence of eIF4A, we observed 66 ± 10% extent of recruitment – which increased to >90% in the presence of eIF4A – comparable to the results seen for the CAA 50-mer in the absence of eIF4A (*Figure 3B*, red vs. black bars, RNA 1 vs. 2). The $k_{max}$ was 3.3 ± 0.4 $min^{-1}$, approximately 2-fold lower than the $k_{max}$ for the CAA 50-mer, whereas the $K_{1/2}^{eIF4A}$ values were indistinguishable for the two mRNAs (*Figure 3—figure supplement 1E*). Importantly, eIF4A strongly stimulated the rate of recruitment of this mRNA (as before, calculated by $k_{max}/k_{obs}^{no\ eIF4A}$); *Figure 3—figure supplement 1E*), in this case by 13-fold (*Figure 3D*, RNA 2). Similar results were obtained for

two additional CAA-repeats 250-mers with the AUG situated 67 or 150 nucleotides from the 5'-end (*Figure 3A*, RNAs 3 and 4). These mRNAs had extents of recruitment of 60–70% in the absence of eIF4A, which increased to >90% in the presence of saturating eIF4A (*Figure 3B*, RNAs 3–4, red vs. black bars). Furthermore, the $k_{max}$ for mRNA recruitment in the presence of saturating eIF4A (*Figure 3C*, RNAs 3–4) were 22- and 7-fold greater than $k_{obs}^{no\ eIF4A}$ (*Figure 3—figure supplement 1E*; *Figure 3D*, RNAs 3–4). The $k_{max}$ and $k_{max}/k_{obs}^{no\ eIF4A}$ values varied by ≤2 fold among RNAs 1–4. The reason for these differences is not clear, but does not seem to correlate with overall mRNA length or number of nucleotides 5' or 3' to the AUG. In summary, all four unstructured CAA-repeats mRNAs that we studied can be recruited by the PIC at appreciable levels independently of eIF4A, but eIF4A still stimulates their rates of recruitment by roughly an order of magnitude.

To probe the effects of defined, stable secondary structures on the functioning of eIF4A in mRNA recruitment, we examined 250-mer mRNAs comprising CAA-repeats throughout the sequence except for a single 21-nt insertion predicted to form a 9 bp hairpin of −10 kcal/mol stability (*Zuker, 2003*), situated in the 5'-UTR either proximal or distal to the 5'-cap (*Figure 3A*, RNAs 5–6, Key; *Supplementary file 1*). Both cap-proximal and cap-distal insertions of this 21-nt sequence into the 5'-UTR of a luciferase reporter conferred strong inhibition of reporter mRNA translation in yeast cells (*Sen et al., 2016*). We confirmed the presence and location of the single hairpin in RNAs 5 and 6 and absence of significant secondary structure in RNA 4 by incubating RNAs 4–6 at 26°C with a 3'−5' RNA exonuclease, ExoT, specific for single-stranded RNA (*Figure 3—figure supplement 2*); (*Deutscher et al., 1984*; *Zeng and Cullen, 2004*).

To our surprise, in the absence of eIF4A, neither the cap-proximal nor the cap-distal hairpins significantly influenced the extent of recruitment, achieving endpoints between 70% and 80%, comparable to the unstructured CAA 50-mer and 250-mer RNAs (*Figure 3B*, RNAs 5–6 vs. 1–4). The $k_{max}$ in the presence of eIF4A for both hairpin-containing mRNAs (RNAs 5–6) were ~2 fold lower than the $k_{max}$ for the CAA 250-mers lacking the hairpin (RNAs 1–4) but the $k_{max}$ values were 2–3-fold higher than the $k_{max}$ for *RPL41A* (*Figure 3C*, RNAs 5–6 vs. 10). The $K_{1/2}^{eIF4A}$ values for RNAs 5 and 6 were 0.20 ± 0.01 μM and 0.10 ± 0.02 μM, respectively. eIF4A stimulated the rate of recruitment ($k_{max}/k_{obs}^{no\ eIF4A}$) for both cap-proximal and cap-distal hairpin mRNAs but, surprisingly, to a lesser degree than in the absence of the hairpin: between 3- and 4-fold as opposed to 7-fold for RNA 4 containing an unstructured 5'-UTR of similar length (*Figure 3D*, RNA 4 vs. 5–6). Thus, at odds with the expectation that stable structures in the 5'-UTR would impose strong obstacles to mRNA recruitment to the PIC, we found that addition of a cap-proximal or cap-distal hairpin in the 5'-UTR of an otherwise unstructured mRNA confers little or no inhibition of the extent of recruitment and only a modest reduction in the rate, in the presence or absence of eIF4A. Moreover, the observation that these hairpins in the 5'-UTR actually decrease the enhancement of the rate of mRNA recruitment provided by eIF4A relative to what we observed with the unstructured mRNA (*Figure 3D*, RNAs 5–6 vs. 4) is not readily consistent with the idea that the factor's predominant function is to unwind stable secondary structures in the 5'-UTRs of mRNAs to facilitate PIC attachment. If this were the case, one might have expected larger rate enhancements for mRNAs with stable structures in their 5'-UTRs than for mRNAs containing little inherent structure, the opposite of what we actually observe. It is possible that eIF4A is not efficient at unwinding such stable structures, which results in a lower $k_{max}$ and a lower degree of stimulation.

To probe further the effects of RNA structural complexity on mRNA recruitment and eIF4A function, we examined a chimeric mRNA comprising CAA-repeats in the 5'-UTR and the natural sequence (with associated structural complexity) from *RPL41A* 3' of the AUG start codon (*Figure 3A*, RNA 7). In the absence of eIF4A, this mRNA was recruited to the PIC with an observed rate ($k_{obs}^{no\ eIF4A}$) of 0.06 ± 0.01 min$^{-1}$, which is significantly slower than for RNAs 1–6 (~0.2–0.9 min$^{-1}$), but faster than the rate for full-length *RPL41A* mRNA, which as noted above could not be determined due to its low endpoint of recruitment (*Figure 3—figure supplement 1B,E* RNAs 1–6, 10 vs. 7). In contrast, in the presence of saturating eIF4A the $k_{max}$ for RNA 7 was comparable to the $k_{max}$ values observed with RNAs 2, 3, 5 and 6 and within 2-fold of the $k_{max}$ values for RNAs 1 and 4. Thus, addition of eIF4A conferred an ~60 fold increase in the rate of recruitment for RNA 7 (i.e., $k_{max}/k_{obs}^{no\ eIF4A}$ = 3.4 min$^{-1}$/0.06 min$^{-1}$ = 57 fold), a greater degree of stimulation than observed with RNAs 1–6 (*Figure 3D*). These results indicate that eIF4A can efficiently resolve inhibition of mRNA recruitment mediated by RNA sequences on the 3' side of the start codon, which are not predicted

to form stable secondary structures with the 5'-UTR (*Figure 1—figure supplement 1D*) (*Zuker, 2003*). In addition, the $K_{1/2}^{eIF4A}$ for RNA 7 (1.8 ± 0.9 µM) was approximately midway between the values determined for the various unstructured model mRNAs and *RPL41A* mRNA (compare RNAs 1–4, 7, and 10 in *Figure 3—figure supplement 1E*), suggesting that structure on the 3' side of the start codon increases the concentration of eIF4A required to maximally stimulate recruitment.

In contrast to the modest effects of the hairpins when present in the otherwise unstructured RNAs 5–6, both the cap-proximal and cap-distal hairpins were strongly inhibitory to mRNA recruitment in the absence of eIF4A when inserted into the unstructured 5'-UTR of the chimeric mRNA harboring *RPL41A* sequence 3' of the AUG codon, conferring very low reaction endpoints (≤20%; *Figure 3B*, RNAs 8–9, red bars) and rates (*Figure 3—figure supplement 1E*, $k_{obs}^{no\ eIF4A}$, RNAs 8–9). Moreover, the presence of saturating eIF4A dramatically increased recruitment of both of these mRNAs to ~80% (*Figure 3B*, RNAs 8–9, black vs. red bars) and also increased their $k_{max}$ values by approximately 10-fold (*Figure 3D*, RNAs 8–9). Thus, the hairpin insertions conferred the predicted strong inhibition of PIC recruitment and marked dependence on eIF4A in the context of the chimeric mRNA containing native *RPL41A* sequences, but not in an otherwise unstructured mRNA. We note however that stimulation of the recruitment rate by eIF4A ($k_{max}/k_{obs}^{no\ eIF4A}$) was considerably less than the ~60 fold and ~190 fold increases observed for the chimeric mRNA lacking a hairpin (RNA 7) and for native *RPL41A* (RNA 10), respectively (*Figure 3D*, RNAs 8–9 vs. 7,10); and the $k_{max}$ values for mRNAs 8–9 also remain well below the values for mRNA 7 and *RPL41A* mRNA (*Figure 3C*, RNAs 8–9 vs. 7 and 10). These differences might be explained by the inability of eIF4A to efficiently resolve these stable secondary structures, as already suggested above.

Comparing $k_{max}$ values for RNA 4 to RNAs 5–9 shows that the combined effects of structures in the 5'-UTR and 3' to the AUG (in RNAs 8 and 9) is considerably more than the additive effects of either structural element alone (in RNAs 5–7) (*Figure 3C*, RNAs 4–9; *Figure 3—figure supplement 1E*). Whereas structures in the 5'-UTR (RNAs 5 and 6) or 3' to the start codon (RNA 7) on their own have 2–3-fold effects on $k_{max}$ relative to the $k_{max}$ for the unstructured RNA 4 (*Figure 3—figure supplement 1E*, compare $k_{max}$ values for RNA 4 and RNAs 5–7: 6.2 min⁻¹ vs. 2.3-3.4 min⁻¹, respectively), combining them (RNAs 8 and 9) produces a ≥ 30 fold effect (*Figure 3—figure supplement 1E*, compare $k_{max}$ values for RNA 4 and RNAs 8 and 9: 6.2 min⁻¹ vs. 0.16 min⁻¹ and 0.20 min⁻¹, respectively). Because the additive effects of structures in the 5'-UTR (RNAs 5 and 6) and structures 3' to the start codon (RNA 7) would only be ~6 fold (~2-fold X ~ 3 fold), the >30 fold effect observed by combining them indicates that structures in both regions synergistically inhibit the rate of mRNA recruitment. This synergy is also reflected in the dramatically reduced recruitment endpoints of RNAs 8–9 seen in the absence of eIF4A (*Figure 3A,B*, red bars; RNAs 4–7 vs. 8–9). It is possible that, in the absence of significant RNA structure on the 3' side of the AUG codon, PICs are able to load directly in the unstructured region containing the AUG codon located downstream of the stem loops in RNAs 5 and 6, in a manner only slightly encumbered by the stable hairpins (*Agalarov et al., 2014*). When the structured *RPL41A* mRNA is present beyond the start codon in RNAs 8 and 9, it may make this direct loading impossible, perhaps by sterically occluding or otherwise blocking access to the internal unstructured region, thereby forcing the PIC to attach near the 5'-cap and scan through the structured 5'-UTR to reach the AUG codon. This latter situation likely reflects the state of most (if not all) natural mRNAs, in which PICs do not have access to large segments of unobstructed, unstructured internal RNA on which to directly load.

The fact that eIF4A restores the extent of recruitment of RNAs 8 and 9 (*Figure 3B*, RNAs 8–9, black bars), albeit at relatively low recruitment rates, indicates that eIF4A can eventually resolve the synergistic inhibition produced by structures in the 5' and 3' segments of these messenger RNAs. It is noteworthy, however, that eIF4A gives considerably larger rate enhancements (*Figure 3D*) for the mRNAs harboring only native *RPL41A* sequences (RNAs 7 and 10) than those burdened with synthetic hairpins (RNAs 5–6 and 8–9), suggesting that eIF4A is better able to resolve the complex array of relatively less stable structures in *RPL41A* compared to a highly stable local structure in the 5'-UTR.

In general, mRNAs with the lowest $k_{obs}^{no\ eIF4A}$ values had the highest $K_{1/2}^{eIF4A}$ values and those with higher $k_{obs}^{no\ eIF4A}$ values had lower $K_{1/2}^{eIF4A}$ values (*Figure 3—figure supplement 1E*, compare RNAs 7–10 to RNAs 1–6). This observation suggests that the $K_{1/2}^{eIF4A}$ values reflect a shift in the rate-limiting step from eIF4A-stimulated mRNA recruitment to a later step (or combination of steps) in the

pathway that is not dependent on eIF4A (*Figure 3—figure supplement 3*). In this model, as the concentration of eIF4A is increased, it decreases the rate-limiting barrier for mRNA recruitment (left hand barriers in *Figure 3—figure supplement 3A,B*) and consequently accelerates the rate of the reaction. Once this barrier is lowered below the level of the next highest barrier (right hand barriers in *Figure 3—figure supplement 3A,B*), no further increase in rate will be observed (i.e., the rate plateaus and apparent saturation is achieved) because the rate becomes limited by this second, eIF4A-independent step. If the eIF4A-dependent barrier is large – for example because the mRNA is highly structured – it takes more eIF4A to get to the point at which the eIF4A-independent step becomes rate-limiting because there is more energetic distance between the two barriers (*Figure 3—figure supplement 3A*). In contrast, if the first barrier is smaller, as expected for mRNAs with little structure, there is less energetic distance between the two barriers and a lower concentration of eIF4A is required to reduce the eIF4A-dependent barrier to the point at which the second, eIF4A-independent barrier becomes rate-limiting, and further increases in eIF4A concentration no longer increase the rate of recruitment. To explain the differences in $k_{max}$ values, we posit that the second, eIF4A-independent barrier is higher for the more structured mRNAs, leading to the observed lower $k_{max}$ values. An alternative model that is also consistent with the data is that even at very high concentrations eIF4A cannot fully reduce the RNA unwinding barrier for the highly structured mRNAs and thus the rate saturates before this barrier reaches the level of the barrier for the eIF4A-independent step (*Figure 3—figure supplement 3C*), leading to low $k_{max}$ values. Even if the situation is more complicated than these simple interpretations, it seems unlikely that the $K_{1/2}^{eIF4A}$ values reflect binding affinities between eIF4A and any one component of the system. Instead, the changes in the $K_{1/2}^{eIF4A}$ values we observe are more likely to be providing information about the relative heights of kinetic barriers along the mRNA recruitment pathway as a function of the degree and distribution of structures in the mRNA.

## The 5′−7-methylguanosine cap imposes an eIF4A requirement for structured and unstructured mRNAs

To further understand the interplay between eIF4A•4G•4E, the PIC and the mRNA during the recruitment process, we inquired how the 5′-cap – which binds the heterotrimer via eIF4E – influences the requirement for eIF4A in recruitment of various RNAs. We have previously shown that the 5′-cap enforces the requirement for several eIFs, including eIF4A, in mRNA recruitment (*Mitchell et al., 2010*) and more recent work in the mammalian system provided evidence that the 5′-cap-eIF4E-eIF4G-eIF3-40S network of interactions is required to promote mRNA recruitment via threading of the 5′-end into the 40S entry channel (*Kumar et al., 2016*). As before, we monitored the kinetics of mRNA recruitment at various concentrations of eIF4A with capped or uncapped versions of mRNAs described above, including RNA 1 (CAA 50-mer), RNA 4 (CAA 250-mer), and RNA 7 (unstructured 5′-UTR with *RPL41A* sequence 3′ of the AUG; *Figure 4* and *Figure 4—figure supplement 1*). As summarized in *Figure 4B*, the $k_{max}$ observed with saturating eIF4A was comparable with or without the 5′-cap for RNAs 1 and 7, and was 1.5-fold lower with the cap than without it for RNA 4. In contrast, in the absence of eIF4A, the rates of recruitment for uncapped versions of the unstructured model mRNAs 1 and 4 were 3.7- and 2.5-fold higher, respectively, than the rates of the corresponding 5′-capped mRNAs (*Figure 4A,B*; *Figure 4—figure supplement 1*, compare 0 μM eIF4A points in the inset graphs). This effect was even more pronounced for RNA 7, containing natural mRNA sequence 3′ of the AUG, which was recruited 15-fold faster when uncapped versus capped in the absence of eIF4A. It is also noteworthy that the rate enhancement provided by eIF4A (*Figure 4B*, $k_{max}/k_{obs}^{no\ eIF4A}$) is larger in all cases for the capped mRNAs than the uncapped mRNAs, reaching an order of magnitude difference for RNA 7, and this effect is due almost entirely to the reduced rate in the absence of eIF4A for the capped versus uncapped mRNAs. Taken together, our data indicate that even for short mRNAs with low structural complexity, the 5′-cap inhibits recruitment in the absence of eIF4A, consistent with our previous observations with a natural mRNA and our proposal that the cap serves, in part, to enforce use of the canonical mRNA recruitment pathway (*Mitchell et al., 2010*). This effect could be due, in part, to the 5′-cap-eIF4E interaction directing the PIC to load at the 5′-end of the mRNA and impeding it from binding directly to downstream, unstructured RNA segments.

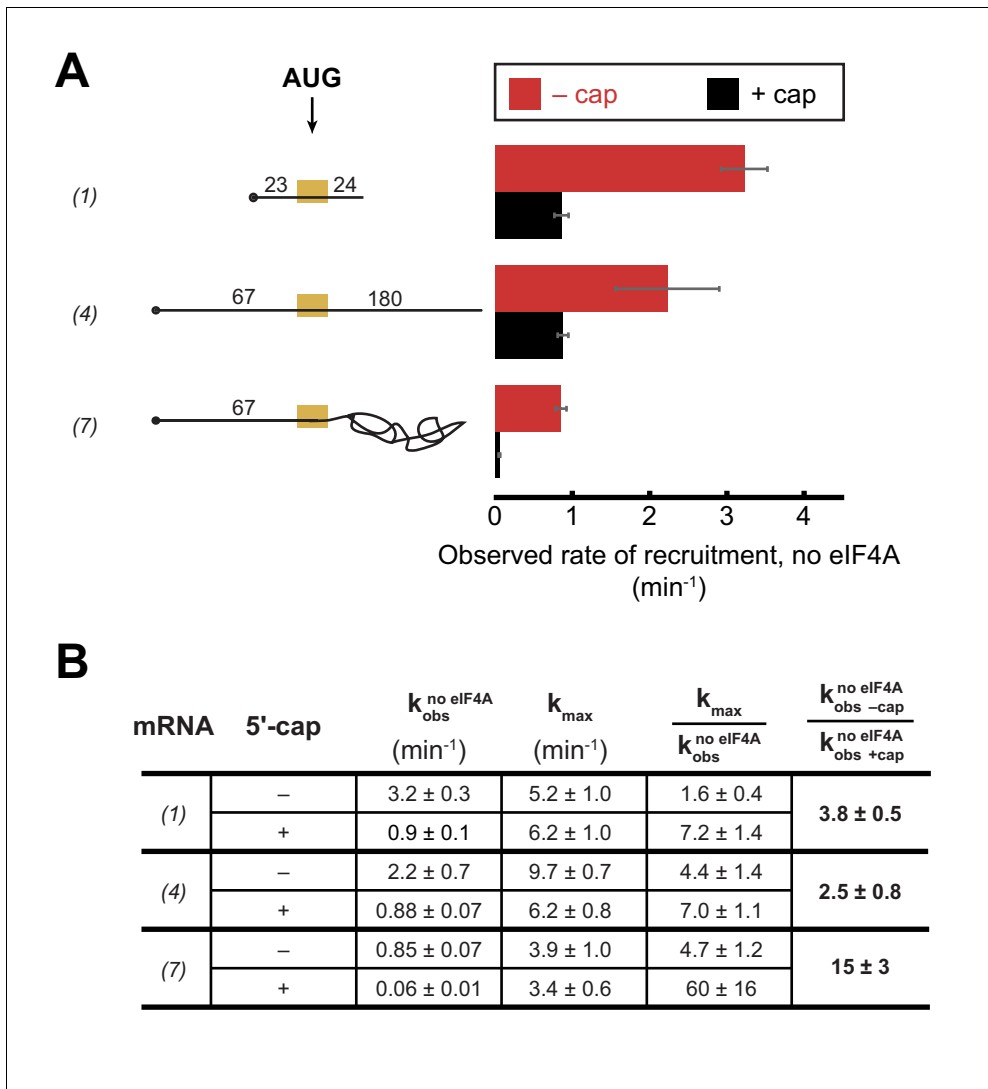

**Figure 4.** The 5'−7-methylguanosine cap inhibits mRNA recruitment in the absence of eIF4A. Observed rates of mRNA recruitment (min$^{-1}$) when eIF4A was not included in the reaction, in the presence (black bars) or absence (red bars) of the 5'-cap. (See *Figure 3A* Key for explanation of mRNA diagrams). (B) The observed rates of recruitment in the absence of eIF4A ($k_{obs}^{no\ eIF4A}$), $k_{max}$, fold enhancement by eIF4A ($k_{max}/k_{obs}^{no\ eIF4A}$), and fold difference in the absence of eIF4A ($k_{obs\ -cap}^{no\ eIF4A}/k_{obs\ +cap}^{no\ eIF4A}$). The data for the capped mRNAs are reproduced from *Figure 3— figure supplement 1E* for comparison purposes. All data presented in the figure are mean values (n ≥ 2) and error bars represent average deviation of the mean. Error was propagated when ratios were calculated.
DOI: https://doi.org/10.7554/eLife.31476.015

The following source data and figure supplement are available for figure 4:

**Source data 1.** Rates of mRNA recruitment measured with capped and uncapped mRNAs in the presence of saturating eIF4A ($k_{max}$) or in the absence of eIF4A ($k_{obs}^{no\ eIF4A}$).
DOI: https://doi.org/10.7554/eLife.31476.017
**Source data 2.** Related to *Figure 4—figure supplement 1*.
DOI: https://doi.org/10.7554/eLife.31476.018
**Figure supplement 1.** The 5'−7-methylguanosine cap inhibits mRNA recruitment in the absence of eIF4A.
DOI: https://doi.org/10.7554/eLife.31476.016

## Discussion

### The PIC stimulates eIF4A and eIF4F ATPase activities independently of eIF4G•4E but dependent on eIF3

In the prevailing model of mRNA recruitment, eIF4F (eIF4A•4G•4E) is localized to the 5'-end of the mRNA via the eIF4E-cap interaction, where it collaborates with eIF4B to unwind structures in the 5'-UTR (*Hinnebusch, 2014*). Given the natural propensity of an mRNA to form structure it is difficult to envision how an mRNA could be unwound by eIF4F and eIF4B, released into the cytoplasm and then bound by the PIC without the mRNA reforming its structure. In another proposed model, eIF4F and eIF4B could interact with the PIC, forming a 'holo-PIC' (*Aitken and Lorsch, 2012*) that relaxes the mRNA and attaches to it synchronously. Some support for a model in which eIF4F interacts with the PIC to promote mRNA recruitment came from hydroxyl radical footprinting experiments that indicated mammalian eIF4G binds to the 40S subunit near the eukaryotic expansion segment 6 of the 18S rRNA (*Yu et al., 2011*). Other work has indicated that yeast eIF4B binds to the 40S ribosomal subunit (*Walker et al., 2013*) and modulates the conformation of the mRNA entry channel, and that mammalian eIF4B interacts with 40S subunit rRNA (*Methot et al., 1996*). These results also support the idea that the eIF4 factors play roles in mRNA recruitment that require binding to the PIC. In addition, in mammals (but not *S. cerevisiae*) eIF4G interacts with eIF3, which could serve to bring the eIF4F complex onto the PIC (*des Georges et al., 2015*). Our observation that the PIC stimulates the ATPase activity of eIF4A, both in the context of the eIF4F complex and in the absence of eIF4G•4E, indicates that eIF4A and eIF4F functionally interact with the PIC, and is consistent with the holo-PIC model. Remarkably, the 6-fold stimulation of eIF4A's ATPase activity by the PIC observed in the absence of eIF4G•4E is comparable to the stimulation provided when eIF4A binds to eIF4G•4E to form the eIF4F complex (4–5-fold; [*Rajagopal et al., 2012*]).

Our data indicate that the Complete PIC is required for full ATPase activation of eIF4A, with the 3g and 3i subunits of eIF3 playing a particularly important role. These eIF3 subunits have been implicated in mRNA recruitment and scanning (*Aitken et al., 2016*; *Cuchalová et al., 2010*; *Valásek, 2012*), and structural data suggest that they are located near the mRNA entry channel of the 40S subunit, on either the solvent or intersubunit face (*Aylett et al., 2015*; *des Georges et al., 2015*; *Llácer et al., 2015*). The observation that these eIF3 subunits appear at distinct locations near the mRNA entry channel in complexes either containing or lacking mRNA has led to the speculation that they might participate in a large-scale rearrangement of the PIC important for either initial attachment to the mRNA or scanning along it (*Llácer et al., 2015*; *Simonetti et al., 2016*). Our observation that the eIF3g and eIF3i subunits are required for full stimulation of eIF4A's ATPase activity is consistent with the possibility that eIF4A is located near the mRNA entry channel, where it can promote mRNA loading onto the PIC (*Marintchev et al., 2009*). Mammalian eIF3 is significantly larger and contains more subunits than the *S. cerevisiae* homolog (*des Georges et al., 2015*); however, both *S. cerevisiae* and mammalian eIF3 contain the core subunits a, b, c, g, and i.

### eIF4A promotes recruitment of all mRNAs regardless of their degree of structure

A number of observations have suggested that the function of eIF4A is to unwind structures in the 5'-UTRs of mRNAs. In vitro, eIF4A can unwind model RNA duplexes (*Andreou and Klostermeier, 2014*; *Blum et al., 1992*; *García-García et al., 2015*; *Grifo et al., 1983*; *Lorsch and Herschlag, 1998b*; *Ray et al., 1985*; *Rogers et al., 1999, 2001*; *Seal et al., 1983*), albeit at a rate slower than necessary to support estimated rates of translation initiation in the range of 10 min$^{-1}$ in vivo (*Palmiter, 1975*; *Shah et al., 2013*; *Siwiak and Zielenkiewicz, 2010*). The factor is required in reconstituted translation systems for 48S PIC formation (*Benne and Hershey, 1978*; *Pestova and Kolupaeva, 2002*; *Schreier and Staehelin, 1973*), and in mammalian extracts the eIF4A-dependence of translation of different reporter mRNAs was correlated with their degree of 5'-UTR structure (*Svitkin et al., 2001*). On the other hand, mammalian eIF4A has also been shown to stimulate initiation in vitro on mRNAs with low degrees of 5'-UTR structure (*Blum et al., 1992*; *Pestova and Kolupaeva, 2002*), and it was shown to perform a critical function in 48S PIC formation on β-globin mRNA that could not be executed by other helicases (Dhx29 or Ded1) more effective than eIF4A in resolving stem-loop structures within 5'-UTRs (*Abaeva et al., 2011*; *Pisareva et al., 2008*). Recent

results indicate that mammalian eIF4A is required for full accommodation of the 5'-UTR of β-globin mRNA in the mRNA entry channel of the 40S subunit (*Sokabe and Fraser, 2017*), and other evidence suggests that eIF4A is required to position eIF4E at the 40S entry channel for threading the capped 5'-end of the mRNA into the 40S subunit (*Kumar et al., 2016*). Ribosome profiling studies in yeast revealed that the vast majority of mRNAs display a similar, strong dependence on eIF4A for efficient translation, whereas mRNAs with long, structured 5'-UTRs generally exhibit a special dependence on the helicase Ded1 in addition to their general requirement for eIF4A (*Sen et al., 2015*). Thus, there is evidence in both mammalian and yeast systems that eIF4A has a general function in PIC attachment to mRNAs while also playing a role for particular mRNAs burdened with 5'-UTR structures. We employed the reconstituted yeast system to test this hypothesis.

We observed that ATP hydrolysis by yeast eIF4A accelerates recruitment of structured as well as unstructured mRNAs, and this acceleration does not correlate with the amount of secondary structure in the 5'-UTR. Taken together with our observation that the PIC accelerates ATP hydrolysis by eIF4A, one explanation for the ability of eIF4A to stimulate mRNA recruitment regardless of 5'-UTR structure would be that one function of eIF4A is to modulate the conformation of the 40S ribosomal subunit, as recently proposed by Sokabe and Fraser (*Sokabe and Fraser, 2017*). For instance, ATP hydrolysis by eIF4A, stimulated at least in part by eIF3 subunits, might act to open the mRNA entry channel, allowing the PIC to engage with and/or move along the mRNA (*Figure 5A*) – similar to the proposed function of mammalian helicase Dhx29 (*Hashem et al., 2013*; *Pisareva et al., 2008*).

## Structural complexity beyond the 5'-UTR inhibits mRNA recruitment in a manner alleviated by eIF4A

We observed that addition of native *RPL41A* mRNA sequence 3' of the start codon inhibits recruitment of mRNA to the PIC, even if the 5'-UTR is made up of CAA-repeats, and this inhibitory effect is ameliorated by eIF4A. This result is surprising because in the canonical model of initiation eIF4A functions to resolve structures in the 5'-UTR. We also found that, in combination, isolated hairpins in the 5'-UTR and structural complexity 3' of the AUG codon synergistically inhibit mRNA recruitment. Thus, structure created by elements beyond the 5'-UTR strongly influences the rate of mRNA recruitment to the PIC and eIF4A is able to diminish these inhibitory effects. We envision that the *RPL41A* sequences 3' of the AUG introduce an ensemble of relatively weak secondary or tertiary interactions that occlude the 5'-UTR and start codon and thus impede direct PIC attachment and AUG recognition, and that eIF4A can efficiently resolve these structural impediments. The *RPL41A* sequences might interact directly with the 5'-UTR or simply create structures that envelop it, sterically occluding access to the 5'-UTR and the start codon. In addition, RNAs longer than their persistence length inherently fold back on themselves (*Chen et al., 2012*), which could further serve to create a steric block to the 5'-UTR. By binding to single-stranded segments, eIF4A may serve to increase their persistence length, helping to untangle the messenger RNA. It is intriguing that eIF4A was 15- to 20-fold more effective in overcoming the inhibitory effect of the *RPL41A* sequences versus the 5'-UTR hairpins in the RNAs containing one or the other inhibitory element (RNA 7 vs. RNAs 5 and 6). This could be explained by proposing that eIF4A is more efficient at resolving the relatively weak interactions that contribute to the overall structure of an mRNA compared to highly-stable local structures. Indeed, only a handful of native mRNAs are known to possess stable hairpin structures in yeast cells (*Rouskin et al., 2014*), yet eIF4A is essential for translation of virtually all mRNAs in vivo. Moreover, it was shown that the DEAD-box RNA helicase Ded1 rather than eIF4A is required to overcome the inhibitory effects of stable hairpin insertions on reporter mRNA translation in vivo (*Sen et al., 2015*). Combined with these previous findings, our results support a model in which yeast eIF4A acts to disrupt moderately stable, transient interactions throughout an mRNA that sequester the mRNA 5'-UTR or start codon within the overall mRNA structure, whereas Ded1 is more effective in resolving stable, local secondary structures in the 5'-UTR or the initiation region of an mRNA.

## A holistic model for eIF4A function in mRNA recruitment

A possible model for eIF4A function that seems consistent with previous studies in both mammalian and yeast systems and the results presented here is shown in *Figure 5B*. This model is not mutually exclusive with that depicted in *Figure 5A* in which eIF4A modulates the structure of the 40S ribosomal subunit. In the holistic model, eIF4A – which is present in large excess of ribosomes in vivo

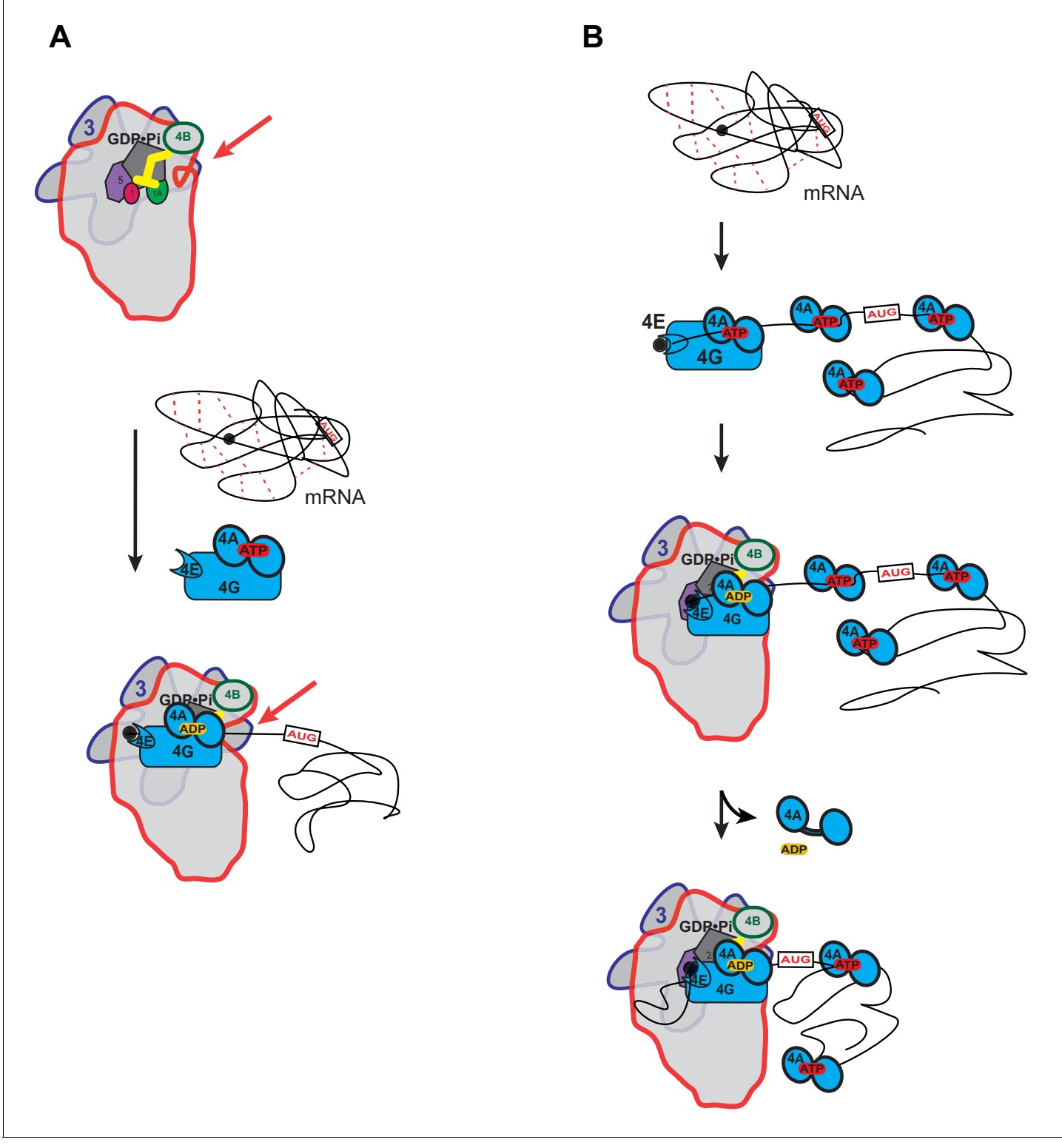

**Figure 5.** Possible models for the mechanism of action of eIF4A in promoting mRNA recruitment to the PIC. (**A**) eIF4A modulates the conformation of the PIC to promote mRNA recruitment. In this model, binding of eIF4A•4G•4E to the PIC induces an opening of the mRNA entry channel of the 40S subunit (highlighted by red arrows), which enables mRNA binding. The ATPase activity of eIF4A might drive this opening. ATP hydrolysis might also promote loading of the message into the mRNA entry channel, concomitant with any necessary melting of structure. This model is consistent with our observations that the ATPase activity of eIF4A is stimulated by the PIC and that recruitment of unstructured mRNAs is accelerated by ATP hydrolysis by the factor. (**B**) A holistic model for the roles of eIF4A and eIF4F in mRNA recruitment. This model attempts to bring a variety of observations from the *Figure 5 continued on next page*

*Figure 5 continued*

literature together with the data presented in this study. mRNAs have complex, often dynamic, structures due to numerous local and distant interactions (red dotted lines) as well as the inherent tendency of polymers longer than their persistence lengths to fold back on themselves. These complex global structures can occlude the 5'-ends of the mRNAs and the start codon, making it difficult for the PIC to bind and locate them. Individual eIF4A molecules might keep the mRNA in a partially-unwound state so that the 5'-end can be located by eIF4F and the PIC. ATP hydrolysis by the eIF4A molecule bound to eIF4F allows it to load the 5'-end of the mRNA into the entry channel of the 40S subunit. Loading might involve opening of the mRNA entry channel or another conformational change in the PIC and active transfer of the mRNA into the channel. After loading of the 5'-end, eIF4A•ADP dissociates from the complex. This allows the next (3') eIF4A molecule on the mRNA to bind to eIF4G, which activates its ATPase, enabling loading of that mRNA segment and thus movement of the PIC down the message. This cycle repeats until the start codon is located. The two models presented in (A) and (B) are not mutually exclusive and it is possible that eIF4A, given its high cellular concentration, performs multiple tasks.
DOI: https://doi.org/10.7554/eLife.31476.019

(*Firczuk et al., 2013*) – binds to an mRNA throughout its length (*Lindqvist et al., 2008*) and mediates the relaxation of local structures, helping to expose the 5'-cap. The eIF4F complex associates with the cap, the 5'-end of the mRNA and the PIC. Direct loading of the PIC onto the 5'-UTR near the start codon is not possible because the start codon is still occluded within the overall structural ensemble of the mRNA and because the cap-eIF4E interaction directs the PIC to the 5'-end. Interaction of the eIF4F complex with the PIC leads to an acceleration of ATP hydrolysis by the bound eIF4A. This hydrolysis event could mediate opening of the mRNA binding channel, as suggested in *Figure 5A*, and loading of the 5'-end of the mRNA into it. It would also result in a low affinity, ADP-bound state of eIF4A that would dissociate from the mRNA, eIF4G and the PIC. During loading of the mRNA, the eIF4E-eIF4G association with the cap may be disrupted, as proposed by Pestova and colleagues (*Kumar et al., 2016*). Movement of the PIC on the mRNA would then lead to its encountering a new eIF4A molecule, which could then associate with eIF4G. Association of the mRNA-bound eIF4A with eIF4G and the PIC would increase its ATP hydrolysis activity by 14-fold, which would again promote mRNA loading into the 40S subunit and dissociation of eIF4A•ADP. This cycle could repeat itself until the start codon is located. The successive cycles of PIC-eIF4G-eIF4A interaction at the entry channel followed by eIF4A dissociation posited in this model could bias the directionality of scanning in the observed 5'−3' direction (*Spirin, 2009*). In addition, the cycling of eIF4A molecules into and out of the complex at each step is consistent with previous observations that an ATPase deficient mutant of eIF4A acts in a dominant-negative fashion to inhibit translation initiation in vitro (*Pause et al., 1994*), as the mutation would block dissociation of eIF4A molecules when they encounter eIF4G in the scanning PIC, and thereby impede progression through the 5'-UTR. Although speculative, this model brings together previous results in the field with our observations that the PIC stimulates ATP hydrolysis by eIF4A and eIF4F, that eIF4A accelerates recruitment of mRNAs regardless of their degree of structure, and that structure throughout the length of mRNAs inhibits recruitment in a manner relieved by eIF4A. It builds upon previous models for eIF4F function (*Yoder-Hill et al., 1993*).

As noted above, there is considerable evidence that mammalian eIF4A can promote PIC attachment and scanning by unwinding stable secondary structures in mRNA 5'-UTRs (*Pestova et al., 1996*; *Svitkin et al., 2001*), and this function might be more prominent in mammalian cells than in yeast. However, even in yeast there is evidence that mRNAs with longer, more structure-prone 5'-UTRs have a heightened requirement for eIF4A, just not to the same extent observed for Ded1 (*Sen et al., 2015*). Thus, in addition to the broader functions depicted in *Figure 5*, eIF4A likely also cooperates with other helicases in resolving stable secondary structures (*Gao et al., 2016*) in 5'-UTRs during PIC attachment and scanning, both in mammals and yeast. The fact that eIF4A is not required for 48S PIC assembly with certain internal ribosome entry sites (IRESs), for example that of hepatitis C virus (*Pestova et al., 1998b*; *Spahn et al., 2001*), can be explained by proposing that IRES interactions with the 40S subunit evoke a conformational change in the ribosome that enables the initiation region of the mRNA to bind directly into the mRNA binding cleft, bypassing all proposed functions of eIF4A in PIC attachment and scanning (*Doudna and Sarnow, 2007*).

Regardless of the model, the events we observe in our mRNA recruitment assay reflect only the first engagement of a PIC with a messenger RNA. Once multiple ribosomes have been loaded onto an mRNA to form a polysome, the structure of the mRNA presumably changes dramatically, and structures on the 3' side of the start codon would play a different role than they do in early rounds

of initiation. Future studies of initiation events on polysomal mRNA might reveal interesting differences from the starting phase of translation.

## Materials and methods

### Materials

ATP, GTP, CTP, and UTP (products 10585, 16800, 14121, 23160, respectively) were purchased from Affymetrix (Santa Clara, CA). [$\alpha$-$^{32}$P]-GTP was from PerkinElmer (product BLU006H250UC) (Waltham, MA). ATP-$\gamma$-S, ADPCP, and ADPNP (products A1388, M7510, and A2647, respectively), S-adenosyl methionine (SAM) (product A7007), and the pyruvate kinase (900–1400 units/mL)/lactate dehydrogenase (600–1000 units/mL) mix from rabbit muscle (product P0294) were from Sigma (St. Louis, MO). NADH disodium salt was from Calbiochem (product 481913) (San Diego, CA). Phosphoenolpyruvate potassium salt was purchased from Chem Impex International, Inc. (product 09711) (Wood Dale, IL). RiboLock RNase inhibitor was from Thermo Fisher Scientific (product EO0381) (Waltham, MA). The RNeasy RNA purification kit was purchased from Qiagen (product 74106) (Germany). Exonuclease T was purchased from New England Biolabs (product M0265S) (Ipswich, MA). The Abnova Small RNA Marker was purchased from Abnova (product number R0007) (Taiwan). SYBR Gold nucleic acid gel stain (product S11494) and Novex 15% TBE-Urea gels (product EC68852BOX) were purchased from Thermo Fisher Scientific. Corning 384-well plates were purchased from VWR International (product 3544) (Radnor, PA).

### Reagent preparation

Eukaryotic initiation factors – eIFs 1, 1A, 2, 3, 4A, 4B, 4E•4G, and 5 – as well as mRNA were prepared as described previously (*Walker et al., 2013*). tRNA$_i$ was charged with methionine as described previously (*Walker and Fredrick, 2008*). Following charging, Met-tRNA$_i$ was separated from contaminating ATP and other nucleotides (left over from the charging reaction) on a 5 mL General Electric (GE) desalting column equilibrated in 30 mM Sodium Acetate (NaOAc), pH 5.5. This step was essential in order to measure the ATP dependence of mRNA recruitment and to accurately control the concentration of ATP in experiments. The Met-tRNA$_i$ and free nucleotide peaks were confirmed with individual standards prepared identically to a charging reaction. Eluted Met-tRNA$_i$ was precipitated with 3 volumes of 100% ethanol at –20°C overnight, pelleted, and resuspended in 30 mM NaOAc, pH 5.5.

### mRNA capping

mRNAs were capped as described previously (*Aitken et al., 2016*). Briefly, RNAs 2–10 at 5 µM were combined with 50 µM GTP, 0.67 µM [$\alpha$-$^{32}$P]-GTP, 100 µM S-adenosyl methionine (SAM), 1 U/µl RiboLock, and 0.15 µM D1/D12 vaccinia virus capping enzyme. RNA 1 was present at 50 µM in the reaction in the presence of 100 µM GTP, with all other conditions identical. Reactions were incubated at 37°C for 90 min and purified using the RNeasy (Qiagen) RNA purification kit.

### mRNA recruitment assay

In vitro mRNA recruitment assays were carried out as described previously with minor modifications (*Aitken et al., 2016*; *Walker et al., 2013*). All reactions were carried out at 26°C in 'Recon' buffer containing 30 mM HEPES-KOH, pH 7.4, 100 mM KOAc, 3 mM Mg(OAc)$_2$, and 2 mM DTT. 15 µl reactions contained final concentrations of 500 nM GDPNP•Mg$^{2+}$, 300 nM eIF2, 300 nM Met-tRNA$_i$, 1 µM eIF1, 1 µM eIF1A, 30 nM 40S ribosomal subunits, 300 nM eIF3, 5 µM eIF4A, 50 nM eIF4G•eIF4E, 300 nM eIF4B, and 1 U/µL Ribolock RNase inhibitor (Thermo). To form the ternary complex (TC), GDPNP and eIF2 were incubated for 10 min, Met-tRNA$_i$ was then added to the reaction and incubated an additional 7 min. The remainder of the components, except mRNA and ATP, were added to the TC and incubated for an additional 10 min to allow complex formation. In all cases when eIF4A was present together with eIF4G•E, it is assumed that the heterotrimeric complex eIF4A•4G•4E (also known as eIF4F) was formed under experimental conditions (*Mitchell et al., 2010*). Reactions were initiated with a mix containing final concentrations of 15 nM mRNA and ATP•Mg$^{2+}$. An alternative incubation scheme in which the mRNA, eIF4A, eIF4G•eIF4E and ATP•Mg$^{2+}$ were preincubated for 2 or 10 min prior to addition of the pre-formed PIC was also tested

in order to determine if unwinding of the mRNA before engagement with the PIC would occur and lead to an increased rate of recruitment. However, this latter incubation scheme resulted in nearly identical $k_{max}$ values as those measured using the scheme described above, with both RPL41A and the CAA 250-mer mRNAs (RNA 4; *Figure 3A*): 1 $min^{-1}$ for RPL41A and 4–6 $min^{-1}$ for the CAA 250-mer. Thus, the first incubation scheme was used in the experiments described in this manuscript.

Experiments varying the concentration of eIF4A were carried out in the presence of 5 mM ATP. Experiments varying ATP were carried out in the presence of 5 μM eIF4A. To take timepoints, 2 μl reaction aliquots were combined with 1 μl of 0.02% bromophenol blue and xylene cyanol dye in 40% sucrose containing a final concentration of a 25-fold excess of unlabeled mRNA (cold chase), identical to the labeled mRNA for that reaction. 2 μl of the chased reaction were immediately loaded and resolved on a native 4% (37.5:1) polyacrylamide gel using a Hoefer SE260 Mighty Small II Deluxe Mini Vertical Electrophoresis Unit at a potential of 200 volts for 50 min. The electrophoresis unit was cooled to 22°C by a circulating water bath. Gels and running buffer contained 34 mM Tris Base, 57 mM HEPES, 0.1 mM EDTA, and 2.5 mM $MgCl_2$ ('THEM'). Gels were exposed to a phosphor plate overnight at –20°C, the plates visualized on a GE Typhoon 9500 FLA, and the fraction of recruited mRNA bands (48S complex) versus the total signal in the lane was quantified using ImageQuant software. Data were plotted and fit using KaleidaGraph 4.5 software. Recruitment time courses were fit to a single exponential rate equation: $y = A*(1-exp(-k_{obs}*t))$, where t is time, A is amplitude, and $k_{obs}$ is the observed rate constant. Observed rates were plotted against the concentration of the titrant and fit to a hyperbolic equation: $y = b + ((k_{max}*x)/(K_{1/2}+x))$ where x is the concentration of the titrant, $k_{max}$ is the maximal observed rate of mRNA recruitment when the reaction is saturated by the factor titrated (e.g., eIF4A), $K_{1/2}$ is the concentration of the factor required to achieve $\frac{1}{2}V_{max}$, and b is the rate in the absence of the titrated factor (i.e., the y-intercept).

## NADH-coupled ATPase assay

The NADH-coupled ATPase assay was adapted from previously described methods with some modifications (*Bradley and De La Cruz, 2012*; *Kiianitsa et al., 2003*). All ATPase experiments were carried out in 384-well Corning 3544 plates on a Tecan Infinite M1000PRO microplate reader at 26°C. Using a standard curve we determined that a 10 μL reaction with 1 mM NADH on a Corning 3544 microplate gives an absorbance of 1.23 Optical Density of 340 nm light ($OD_{340}$) in the microplate reader. $OD_{340}$ was measured every 20 s for 40 min, plotted vs. time for individual reactions, and fit to $y = mx + b$ where m is the slope, x is time in min, and b is the y-intercept. Thus, m is $OD_{340}$ of NADH/min. It follows that,

$$\frac{|m| \text{ OD of NADH/min}}{1.23 \text{ OD of NADH/1mM NADH}} = mM \, NADH/min$$

Note that the absolute value of m ($|m|$) was used because the slope is a negative value due to loss of absorbance over time. NADH consumed is stoichiometric with ATP regenerated thus, mM NADH/min = mM ATP/min. $k_{cat}$ was determined by dividing the velocity of ATP by enzyme (5 μM eIF4A) concentration (*Fersht, 1999*).

12 μl reactions (final volume) were assembled at final concentrations as follows: 1 mM GDPNP•$Mg^{2+}$, 500 nM eIF2, 500 nM Met-tRNA$_i$, 1 μM eIF1, 1 μM eIF1A, 500 nM 40S ribosomal subunits, 500 nM eIF3, 5 μM eIF4A, 500 nM eIF4G•4E, 500 nM eIF4B, 500 nM eIF5, 5 mM ATP•$Mg^{2+}$, 500 nM mRNA, and 1 U/μl RiboLock RNase inhibitor. In all cases when eIF4A was present together with eIF4G•4E, it is assumed that the heterotrimeric complex eIF4A•4G•4E (also known as eIF4F) was formed under experimental conditions (*Mitchell et al., 2010*). All incubations and experiments were performed at 26°C. PICs were formed at 2x of the final concentration in 1x Recon buffer, in the absence of mRNA and ATP. PIC formation was initiated by incubating eIF2 and GDPNP for 10 min, subsequently Met-tRNA$_i$ was added to the reaction and incubated for another 7 min. Next, eIFs 1, 1A, 40S, 3, 5, 4A, 4B, 4G•4E, and RiboLock were added. Order of addition of eIFs did not make a difference. The reaction was incubated for 10 min to allow complex formation. Subsequently, the PICs were combined with the 'Reporter Mix' containing phosphoenolpyruvate, NADH, pyruvate kinase, and lactate dehydrogenase (added as a 10x stock of the final concentrations all in 1x Recon buffer) resulting in concentrations in the final reaction of 2.5 mM phosphoenolpyruvate, 1 mM NADH, and a 1/250 dilution of the pyruvate kinase (600–1,000 units/mL) and lactate dehydrogenase

(900–1400 units/mL) mix (PK/LDH mix). Reactions were brought up to volume with 1x Recon buffer, such that when they were initiated by addition of mRNA (added as a 10x stock of the final concentration in 1x Recon) and ATP•Mg$^{2+}$ (added as a 4x stock of the final concentration in 1x Recon buffer) the total reaction volume was 12 µl. 10 µl of the reaction were then immediately transferred to the microplate for analysis by the Tecan plate reader and changes in absorbance of 340 nm light were monitored over time, taking readings every 20 s for 40 min. Because the assay measures the slope of a straight line (i.e., multiple turnover conditions), the lag in time between initiating the reaction and start of the measurements by the plate reader has no effect on the results. Initiating the reactions using an injection, capable of monitoring rapid kinetics did not reveal any differences in results; that is, there was no evidence of a 'burst' phase in the initial part of the reaction. Increasing or decreasing the concentration of the PK/LDH mix by 3-fold did not influence the observed rate of ATP hydrolysis (*Figure 2—figure supplement 1A*), indicating that the rate of NADH oxidation is not limited by PK/LDH activity. Also, when ATP, eIF4A, or PK/LDH was absent from the reaction, there was no change in absorbance at 340 nm over 1 hr (*Figure 2—figure supplement 1A*).

## Exonuclease T RNA digest

In a 20 µl reaction 0.5 pmol/µl of RNA was incubated with 0.75 U/µl of RNase Exonuclease T in 1x NEB buffer 4 at 26°C for 18 hr. RNA (4 pmol total RNA per lane) was loaded and resolved on a Novex 15% Tris Borate EDTA Urea gel and stained with SYBR Gold nucleic acid gel stain, diluted 1/10,000, for 5 min and visualized on a General Electric Typhoon FLA 9500.

## Acknowledgements

The authors would like to acknowledge Tom Dever and Nicholas Guydosh for their thoughtful and critical suggestions. This work was supported by the Intramural Research Program (JRL and AGH) of the National Institutes of Health (NIH).

# Additional information

## Competing interests

Alan G Hinnebusch: Reviewing editor, *eLife*. The other authors declare that no competing interests exist.

## Funding

| Funder | Author |
| --- | --- |
| National Institutes of Health | Paul Yourik<br>Colin Echeverría Aitken<br>Fujun Zhou<br>Neha Gupta<br>Alan G Hinnebusch<br>Jon R Lorsch |

The funders had no role in study design, data collection and interpretation, or the decision to submit the work for publication.

## Author contributions

Paul Yourik, Conceptualization, Data curation, Formal analysis, Investigation, Visualization, Methodology, Writing—original draft, Writing—review and editing; Colin Echeverría Aitken, Conceptualization, Resources, Formal analysis, Supervision, Visualization, Writing—review and editing; Fujun Zhou, Resources; Neha Gupta, Conceptualization, Resources, Writing—review and editing; Alan G Hinnebusch, Conceptualization, Formal analysis, Supervision, Writing—review and editing; Jon R Lorsch, Conceptualization, Resources, Formal analysis, Supervision, Funding acquisition, Visualization, Project administration, Writing—review and editing

## Author ORCIDs

Paul Yourik (iD) http://orcid.org/0000-0003-0073-1623
Colin Echeverría Aitken (iD) https://orcid.org/0000-0003-2187-0614
Alan G Hinnebusch (iD) http://orcid.org/0000-0002-1627-8395
Jon R Lorsch (iD) http://orcid.org/0000-0002-4521-4999

## Decision letter and Author response

Decision letter https://doi.org/10.7554/eLife.31476.023
Author response https://doi.org/10.7554/eLife.31476.024

## Additional files

### Supplementary files

• Supplementary file 1. RNAs used in the study.
DOI: https://doi.org/10.7554/eLife.31476.020

• Transparent reporting form
DOI: https://doi.org/10.7554/eLife.31476.021

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
