## [Decision Letter]

[Editors’ note: a previous version of this study was rejected after peer review, but the authors submitted for reconsideration. The first decision letter after peer review is shown below.]

Thank you for submitting your work entitled "eIF4A is stimulated by the pre-initiation complex and enhances recruitment of mRNAs regardless of structural complexity" for consideration by *eLife*. Your article has been reviewed by three peer reviewers, and the evaluation has been overseen by a Reviewing Editor and a Senior Editor. The reviewers have opted to remain anonymous.

Our decision has been reached after consultation between the reviewers. Based on these discussions and the individual reviews below, we regret to inform you that your work will not be considered further for publication in *eLife*.

As you will see, the reviewers thought that the question was interesting, but unfortunately, they also thought that the data were not compelling, with complementary concerns. For example, reviewer 1 finds that the evidence in support of the two main conclusions is insufficient, while reviewer 3 agrees that "there is in fact little direct supporting data in the manuscript for the model" presented in Figure 5. In addition, various significant specific concerns were raised. Given these assessments, we have no choice but to decline the manuscript.

*Reviewer #1:*

The paper addresses the role of an ATPase eIF4A in mRNA recruitment. The authors make two main statements: (i) PIC stimulates the ATPase of eIF4A, indicating that the factor works in association with the ribosome complex rather than exclusively with mRNA and (ii) ATP hydrolysis by eIF4A accelerates the rate of mRNA recruitment regardless of the mRNA structure in the 5'UTR. I cannot see sufficient evidence in support of either statement as detailed below.

1) The authors measure multiple turnover ATPase activity of eIF4A, which turns out to be hardly stimulated by the mRNA (30 µM/min without and 40 µM/min with mRNA in Figure 2). The functional relevance of this activity remains unclear and it is difficult to compare it with the rate of translation initiation. Related to this problem, the rate of the multiple-turnover ATPase monitored in the paper is rather low. How does it correlate to the rate of translation initiation in vivo? In this context, the authors should refer to the newer work that allows to calculate the rate of initiation based on the ribosome profiling experiments, rather than citing a paper from 1975 (Introduction, sixth paragraph). The mRNA-independent multiple-turnover activity is not explained by the model presented in Figure 5 and seems inconsistent with the importance of eIF4A for mRNA recruitment (Figure 3 and Figure 4 and existing literature).

2) The only reliable effects on the ATPase are on k_cat_ by 40S and on K_m_ by 4G-4E. Other effects of factor omission are small and can be hardly interpreted. The paper should be shortened considerably when describing these effects.

3) In Figure 3 and Figure 4, the authors present titrations by eIF4A to study mRNA recruitment. This makes it very difficult to compare with the ATPase experiments, because eIF4A is an enzyme in the ATPase reaction, and thus the dependence on eIF4A for the ATPase rate should be linear. Instead, for the mRNA recruitment it is an "activator", hence the hyperbolic dependence. The authors do not provide sufficient kinetic link between the ATPase and mRNA recruitment. Exactly this is an important point which is not directly shown and remains unclear and is also not explained by the model of Figure 5.

4) Except for the tightly folded hairpin structures, there is no evidence that the mRNAs used in this paper form any secondary or tertiary structures or form any sort of collapsed polymer chains. Thus, the statements concerning the role of eIF4A in unfolding such structures (or stabilizing the unfolded state) are speculations not related to the data.

Technical problems:

1) Figure 1: the data are not well presented, as the initial phase of kinetic experiments are compressed at the Y axis. Use a log time axis or at least allow for some distance between the Y axis and the beginning of the time courses. For the faster cases, the rate constant appears to rely on a single point, because the end level is achieved after 2-3 min (again, this is not well presented). Add more points at the initial part of the curve. Important: show the time courses of mRNA recruitment and of ATP hydrolysis in the same time course. Minor problem: Dark blue and black symbols are difficult to distinguish, use distinct colors.

2) In Figure 2, It remains unclear whether "+" in A, B, C, E is with or without mRNA, because the values all differ, e.g., 30 µM/min without mRNA and 40 µM/min with mRNA in A, 14 µM/min for PIC (without mRNA?) in B and 40 µM/min for PIC in C (with mRNA as in A?). This has to be clearly stated and the values have to be comparable. The experiment shown in Figure 2 is kinetically senseless, as it represents an ATPase measurement with a mixture of PIC-stimulated and unstimulated reactions; no reliable information can be extracted here.

3) Figure 4—figure supplement 1. The difference between (7) -cap and (7) +cap relies on one experimental point at low concentration of eIF4A; this is not convincing, more points should be added.

4) The paper is exceedingly long, with significant redundancy between the Introduction and Discussion, and is not well written.

*Reviewer #3:*

Here Yourik et al. use a yeast in vitro reconstituted system to demonstrate that mRNA recruitment to the PIC is stimulated by eIF4A regardless of degree of secondary structure. They also find that global structure per se rather than structure located within the mRNA 5'UTR confers a requirement for eIF4A. The experiments appear well performed and the manuscript is well written. My main concern is that an appropriate explanation for the effects seen by the authors (That is, why does global mRNA structure [even downstream of the AUG codon]) (and the 3'UTR?) impart eIF4A responsiveness? The authors do present a working model to explain why this could be so (Figure 5), but there is in fact little direct supporting data in the manuscript for this model. Does eIF4A bind to the coding region before ribosome recruitment? How does binding of eIF4A to the mRNA body stimulate 5' end-mediated initiation events? As such we are left with a real interesting observation, but no understanding into how it could be functioning. If eIF4A is required to resolve global structure, then what makes TISU and the HCV mRNA (or chimeric templates containing these 5'UTR) eIF4A-independent? I also assume that the mechanism presented in Figure 5 will only be valid for the first round of translation initiation, since thereafter the mRNA will be translating with most internal structure distorted by the 80S ribosomes. Hence eIF4A-mediated unwinding of global structure should not be required for subsequent rounds of re-initiation.

How do the authors reconcile their data with those of Christopher Fraser's group where substantial initiation events were observed on (CAA)4-5'UTR betaglobin-β-globin mRNA (PMID 28559306, Figure 2) in the absence of eIF4A?

[Editors’ note: what now follows is the decision letter after the authors submitted for further consideration.]

Thank you for submitting your article "eIF4A is stimulated by the pre-initiation complex and enhances recruitment of mRNAs regardless of structural complexity" for consideration by *eLife*. Your article has been reviewed by three peer reviewers, and the evaluation has been overseen by a Reviewing Editor and Detlef Weigel as the Senior Editor. The reviewers have opted to remain anonymous.

The reviewers have discussed the reviews with one another and the Reviewing Editor has drafted this decision to help you prepare a revised submission.

Summary:

This is a revised version of a manuscript in which the authors study the mechanism of action of eIF4A in mRNA recruitment using an in vitro reconstituted translation initiation system from budding yeast and mRNAs with varying degrees of structure. They present novel data showing that the PIC stimulates the ATPase activity of eIF4A. ATP hydrolysis by eIF4A accelerates the rate of recruitment for all mRNAs that were employed, regardless of their degree of secondary structure. They conclude that eIF4A also exhibits an important general role in loading mRNAs onto the PIC, in addition, unwinding mRNA structure. The authors conclude that eIF4A breaks many weak interactions formed within a mRNA that occludes the 5'-UTR and start codon and then acts to facilitate engagement of the 5'-cap structure. The revised manuscript is significantly improved over the original version. The data regarding transcripts #4 vs. #7 are interesting and provocative – enough so, to allow the speculation and the revised model in Figure 5 (even though the initiation of HCV and TISU mRNAs remain unexplained).

Essential revisions:

1) While many of the experiments are insightful, the ability of the PIC to stimulate the eIF4A-dependent hydrolysis of ATP is less satisfactory, and should perhaps be dropped from the title. Also, because the experiments were performed with yeast eIF4A, this should be indicated in the title, if it is not changed: "Yeast eIF4A is stimulated by the pre-initiation complex and enhances recruitment of mRNAs regardless of structural complexity".

2) None of the models explain the difference in the affinity of eIF4A to the PIC in the presence of different mRNAs. If eIF4A is recruited through the eIF4F complex to the cap and moves towards the start codon or is required to change the conformation of the 40S, its affinity is defined by the interactions with the 4F complex or the ribosome. It is not clear how the mRNA concentration should enter here and this should be explained.

3) Subsection “eIF4A relieves inhibition of recruitment produced by structures throughout the length of mRNAs”, ninth paragraph. The >30-fold effect by a combination of secondary structures in the 5'UTR and the coding region is not obvious, as it looks like 9-11-fold. Furthermore, even an additive effect (if translated in terms of deltaG) should lead to 3x60 (xfold for construct 6 multiplied by xfold for construct 7) = 180-fold acceleration, compared measured 11 or 9 for mRNAs 8 and 9. Something does not add up here.

4) Introduction – "[…]no apparent sequence specificity[…]" As noted in the recent publication from the Ingolia lab (Iwaski et al., 2016), there does appear to be a preference for a polypurine sequence.

Also in the Introduction – "[…]ATP hydrolysis causes a decrease in affinity of eIF4A for RNA[…]" It would seem that eIF4A•ADP does not bind RNA well, however, the slow chemical step of ATP hydrolysis is associated with tighter binding of RNA as evidenced by either nitrocellulose filter binding or UV crosslinking.

5) Concerning the in vivo rate of initiation – it is likely that all in vitro studies that examine only the binding of a single ribosome to a mRNA will likely be an order of magnitude slower than the rate measured in vivo as the in vivo rate could reflect that of polysomal re-initiation, which is much more efficient (that would be a consequence of an intramolecular effect).

6) The stimulation of ATP hydrolysis is about 2-fold by the 43S/PIC (Figure 2). This is not very dramatic given the number of components involved, the presence of RNA in the 40S subunit and the presence of components known to enhance eiF4A function (eIF4G/4E and eIF4B). In fact, the limited stimulation seen with added eIF4G/4E may readily reflect that it is present at only 10% of the concentration of eIF4A. Perhaps the most unusual feature of the data in Figure 2, which is the apparent inhibition by either eIF4B or eIF5. Secondly, as noted in panel B of Figure 2, roughly 75% of the ATPase activity of the combination of the 40S/PIC + eIF4A occurs in the absence of added RNA.

7) Figure 2—figure supplement 1 – previous studies (see also Lorsch and Herschlag) have indicated an eIF4A Km value for ATP in the 200-400 nM range. What is so different to cause a change such as that seen in panel D?

8) Discussion – an extensive discussion is centered around a limited stimulation of the eIF4A-dependent ATP hydrolysis. This should be considerably condensed. A clear contrast of yeast vs. mammalian systems should be highlighted.

---

## [Author Response]

[Editors’ note: the author responses to the first round of peer review follow.]

Reviewer #1:The paper addresses the role of an ATPase eIF4A in mRNA recruitment. The authors make two main statements: (i) PIC stimulates the ATPase of eIF4A, indicating that the factor works in association with the ribosome complex rather than exclusively with mRNA and (ii) ATP hydrolysis by eIF4A accelerates the rate of mRNA recruitment regardless of the mRNA structure in the 5'UTR. I cannot see sufficient evidence in support of either statement as detailed below.1) The authors measure multiple turnover ATPase activity of eIF4A, which turns out to be hardly stimulated by the mRNA (30 µM/min without and 40 µM/min with mRNA in Figure 2). The functional relevance of this activity remains unclear and it is difficult to compare it with the rate of translation initiation. Related to this problem, the rate of the multiple-turnover ATPase monitored in the paper is rather low. How does it correlate to the rate of translation initiation in vivo? In this context, the authors should refer to the newer work that allows to calculate the rate of initiation based on the ribosome profiling experiments, rather than citing a paper from 1975 (Introduction, sixth paragraph). The mRNA-independent multiple-turnover activity is not explained by the model presented in Figure 5 and seems inconsistent with the importance of eIF4A for mRNA recruitment (Figure 3 and Figure 4 and existing literature).

We agree that the reasons for our choice of multiple-turnover kinetic experiments, in lieu of single-turnover kinetics experiments, to study the ATPase activity of eIF4A and eIF4F were not made clear. Due to technical limitations we cannot measure eIF4A ATPase activity under single turnover conditions as we did for mRNA recruitment. We have now added a section to the text describing the reasons we used multiple-turnover kinetics and making clear the inherent strengths and weaknesses of this approach (subsection “The ATPase activity of eIF4A•4G•4E is increased by the PIC”, first paragraph). Despite the limitations, measuring the ATPase kinetics under multiple turnover conditions provides important insights into if and how eIF4A, eIF4G•4E, and the rest of the PIC functionally interact. Although burst phase kinetics cannot be observed, our results clearly demonstrate that eIF4A’s ATPase activity is stimulated by the PIC in both the presence and absence of mRNA, as described in more detail below.

We also agree with the reviewer that more modern estimates of initiation rates would be desirable additions to the older value we cited. We have been unable to find direct experimental measurements of initiation rates using ribosome profiling. We consulted with several experts in this area – Nick Guydosh (NIDDK, NIH) and Nick Ingolia (UC Berkeley) – and both indicated that they were unaware of the use of ribosome profiling to measure initiation rates. Although no direct measurement of the rate of translation initiation using ribosome profiling appears to be available, we have added citations of recent computational modeling studies that utilize ribosome profiling data as one component of overall estimates of quantitative parameters for protein synthesis. If we have missed any relevant studies in our search, we would be happy to include them.

The values of k_cat_ for ATP hydrolysis by the eIF4A•4G•4E complex in the presence of the PIC (~8 min^-1^) are well within the estimated range of the rates of translation initiation in vivo (e.g., 110 min^-1^).

Lastly, we have completely reworked the explanation of how our data fit into possible models for the mechanism of mRNA recruitment (e.g., subsection “A holistic model for eIF4A function in mRNA recruitment”) and have revised the final figure (Figure 5) accordingly. We also point out that our data are consistent with a recent proposal by Sokabe and Fraser that eIF4A may act, in part, to modulate the conformation of the PIC – for instance, the opening and closing of the mRNA entry channel in the 40S subunit – facilitating loading of mRNAs regardless of their degree of structure. We have modified Figure 5 to contain such a model as one of the possible interpretations. This idea, which is not mutually exclusive with the holistic model presented in Figure 5, is consistent with our observations that eIF4A accelerates the rate of recruitment even for unstructured mRNAs and that the PIC stimulates the rate of ATP hydrolysis by the factor, indicating a functional interaction between them.

2) The only reliable effects on the ATPase are on k_cat_ by 40S and on K_m_ by 4G-4E. Other effects of factor omission are small and can be hardly interpreted. The paper should be shortened considerably when describing these effects.

Our original presentation of activation of eIF4A’s ATPase by the PIC and its components was confusing and we have now substantially reworked it, both in the text and in Figure 2.

The presence of the PIC accelerates ATP hydrolysis by 4-fold for the eIF4A•4G•4E heterotrimer (Figure 2) and by 6-fold for eIF4A alone (Figure 2) compared to the rates observed in the absence of the PIC. These effects are similar in magnitude to the previously established stimulation of eIF4A’s ATPase by eIF4G•4E. Thus, our data suggest that there is a functional interaction between eIF4A and the PIC that is comparable in efficacy to the canonical interaction of eIF4A with the other two subunits of eIF4F. Our data also show that the eIF3g and eIF3i subunits of the eIF3 complex are fully required for this stimulation, as is an intact 43S PIC.

For clarity, we revised Figure 2 to report k_cat_ values (min^-1^), which are much easier to compare to observed rate constants reported elsewhere in the paper and in the literature than the velocity values (µM/min) we presented originally. In order to provide an easier interpretation of the ATPase data we have also added a color code and legend to Figure 2 and moved the K_m_ values to Figure 2—figure supplement 1. We also changed Figure 2 panels C and D (blue and pink cross-hatched bars) to emphasize that omission of eIF2, eIF3, or eIF4G•4E had comparable or even greater effects on the k_cat_ than did omitting the 40S subunit (Figure 2, compare "Complete PIC" to "-40S", "-2", "-3" and Figure 2 "-eIF4G•4E"). As suggested by reviewer 1, we significantly shortened the ATPase section, removed panels B and D from Figure 2, and included two tables in Figure 2—figure supplement 1 that display all of the values for a more facile comparison of the k_cat_ and K_m_ effects.

3) In Figure 3 and Figure 4, the authors present titrations by eIF4A to study mRNA recruitment. This makes it very difficult to compare with the ATPase experiments, because eIF4A is an enzyme in the ATPase reaction, and thus the dependence on eIF4A for the ATPase rate should be linear. Instead, for the mRNA recruitment it is an "activator", hence the hyperbolic dependence. The authors do not provide sufficient kinetic link between the ATPase and mRNA recruitment. Exactly this is an important point which is not directly shown and remains unclear and is also not explained by the model of Figure 5.

We titrated eIF4A in the mRNA recruitment experiments in order to measure the concentration of the factor required to achieve one-half of the maximal rate (K_1/2_). This also allows us to measure the maximal observed rate constant for recruitment (k_max_), which is what we report in Figure 3. Changes in K_1/2_ values can suggest mechanistically important differences in eIF4Adependent mRNA recruitment efficiency and thus we felt this was an important parameter to measure and, in some cases, discuss. We agree with the reviewer that we did not adequately explain the connection (and caveats) between the ATPase and mRNA recruitment experiments, and, as explained in response #1 above, we have now attempted to do so. Also as explained above, we have significantly revised our description of how our data meshes with previous studies and suggests a holistic model for the role of eIF4A in mRNA recruitment.

4) Except for the tightly folded hairpin structures, there is no evidence that the mRNAs used in this paper form any secondary or tertiary structures or form any sort of collapsed polymer chains. Thus, the statements concerning the role of eIF4A in unfolding such structures (or stabilizing the unfolded state) are speculations not related to the data.

To address this concern, we have now included structures of RNAs 1, 7, and 10 as predicted by mfold – a widely used tool in the field for structure prediction – supporting the contention that RNA 1 has a minimal degree of structure, RNA 7 has structure but only beyond the start codon, and RNA 10 has structure throughout the sequence (Figure 1—figure supplement 1). Moreover, there is a large body of work indicating that nearly all natural sequences have a tendency to form structure via numerous interactions (reviewed in Halder and Bhattacharyya, 2013).

Technical problems:1) Figure 1: the data are not well presented, as the initial phase of kinetic experiments are compressed at the Y axis. Use a log time axis or at least allow for some distance between the Y axis and the beginning of the time courses. For the faster cases, the rate constant appears to rely on a single point, because the end level is achieved after 2-3 min (again, this is not well presented). Add more points at the initial part of the curve. Important: show the time courses of mRNA recruitment and of ATP hydrolysis in the same time course. Minor problem: Dark blue and black symbols are difficult to distinguish, use distinct colors.

We have modified Figure 1 plots to show the early time points by moving the y-axis. We also added an inset to Figure 2, which shows 3 time-points in the first minute (earliest at 20 sec.) of the reaction, which is the fastest that we could measure. For the CAA-50mer the beginning of the curve is not covered well, as the reviewer suggests, thus the observed rate (4.0 min^-1^) is an estimate; however, because the reaction achieves a higher degree of mRNA recruited over time than do the reactions lacking either eIF4A or ATP the conclusion still holds that ATP and eIF4A both enhance recruitment of the CAA 50-mer. We now make clear that the 4 min^-1^ rate constant is an estimate (subsection “ATP hydrolysis by eIF4A promotes recruitment of the natural mRNA *RPL41A* as well as a short, unstructured model mRNA”, third paragraph).

As explained above, for technical reasons ATPase kinetics were monitored under multiple turnover conditions whereas recruitment could be studied in the single-turnover regime. Thus, we cannot directly compare them side-by-side. However, the changes in the rate of ATP hydrolysis provide insights into how the eIF4A ATPase activity is affected by the presence of the PIC, both in the presence and absence of mRNA.

As recommended, we made the blue data points in the plots a lighter color so that they are easier to distinguish from the black.

2) In Figure 2, It remains unclear whether "+" in A, B, C, E is with or without mRNA, because the values all differ, e.g. 30 µM/min without mRNA and 40 µM/min with mRNA in A, 14 µM/min for PIC (without mRNA?) in B and 40 µM/min for PIC in C (with mRNA as in A?). This has to be clearly stated and the values have to be comparable. The experiment shown in Figure 2 is kinetically senseless, as it represents an ATPase measurement with a mixture of PIC-stimulated and unstimulated reactions; no reliable information can be extracted here.

We have completely reformatted Figure 2, including by adding labels clearly indicating where there is a saturating amount of ATP or *RPL41A* mRNA: panels A and B have a saturating amount of ATP while *RPL41A* mRNA is varied and panels C and D have a saturating amount of *RPL41A* while ATP is varied. We also added a legend and color-coded Figure 2. Figure 2 in the previous manuscript showed normalized fold stimulation, which is the reason the numbers did not match, but as this panel was not essential for our arguments and was confusing, we have omitted it for clarity. Lastly, as suggested by reviewer 1, we have omitted Figure 2 and the corresponding text, thus shortening the ATPase section further.

3) Figure 4—figure supplement 1. The difference between (7) -cap and (7) +cap relies on one experimental point at low concentration of eIF4A; this is not convincing, more points should be added.

Reviewer 1 correctly points out that the difference in the experimental data is in fact at low concentrations of eIF4A and especially at no eIF4A (i.e., the y-intercept). Our intent is to demonstrate that the main difference in the observed rates, plotted as a function of eIF4A concentration, occurs in the absence of eIF4A. To illustrate this point more clearly, we added insets with a magnified view of the observed rates at low concentrations of eIF4A, up to 0.5 µM.

4) The paper is exceedingly long, with significant redundancy between the Introduction and Discussion, and is not well written.

We rewrote and shortened large portions of the manuscript, with particular attention to the ATPase section in the Results and minimizing redundancy between the Introduction and the Discussion sections.

Reviewer #3:Here Yourik et al. use a yeast in vitro reconstituted system to demonstrate that mRNA recruitment to the PIC is stimulated by eIF4A regardless of degree of secondary structure. They also find that global structure per se rather than structure located within the mRNA 5'UTR confers a requirement for eIF4A. The experiments appear well performed and the manuscript is well written. My main concern is that an appropriate explanation for the effects seen by the authors (That is, why does global mRNA structure [even downstream of the AUG codon]) (and the 3'UTR?) impart eIF4A responsiveness? The authors do present a working model to explain why this could be so (Figure 5), but there is in fact little direct supporting data in the manuscript for this model.

We appreciate the concern of reviewer 3 and we revised our manuscript and Figure 5 to convey better how our data integrate with previous studies to suggest a holistic model for the role of eIF4A, as described in response #1 to reviewer 1. We note that our data are also consistent with a recent model suggesting that eIF4A modulates the conformation of the 40S ribosomal subunit to allow mRNA loading (Sokabe and Fraser, 2017). In this model, modulation of the conformation of the ribosome by eIF4A would stimulate the recruitment of any mRNA. We formatted Figure 5 accordingly, adding an illustration of such a model. We further note that the two models presented in Figure 5 are not mutually exclusive.

Does eIF4A bind to the coding region before ribosome recruitment? How does binding of eIF4A to the mRNA body stimulate 5' end-mediated initiation events? As such we are left with a real interesting observation, but no understanding into how it could be functioning.

Previous work strongly suggests that eIF4A does not have any sequence specificity and binds throughout the length of the mRNA (Lindqvist et al., 2008; Rajagopal et al., 2012). Given the high concentration of eIF4A in the cell (Firczuk et al., 2013) we believe that a reasonable inference would be that eIF4A can bind to an mRNA prior to translation initiation. mRNAs form numerous intramolecular interactions (Halder and Bhattacharyya, 2013), which may lead to occluding the 5'-end and start codon by sequences located throughout the rest of the message. We cannot provide any additional mechanistic insights at this time but believe that the previous body of work allows us to speculate that eIF4A binding to the message would relax the overall folded structure in order to make the 5’-end accessible to binding by eIF4F (Figure 5) as suggested previously (Spirin, 2009).

If eIF4A is required to resolve global structure, then what makes TISU and the HCV mRNA (or chimeric templates containing these 5'UTR) eIF4A-independent?

This is an interesting point. We suspect that these elements allow alternative modes of entry, perhaps, in part, by making accessible binding sites for certain components of the translational machinery in a manner that escapes occlusion by the overall fold of an mRNA. For example, an IRES may be large enough to dominate the structural landscape of a region of an mRNA. Like IRESs, TISU elements appear to allow mRNA recruitment in a manner distinct from canonical translation initiation. Perhaps part of the function of the TISU element is to ensure the 5’-end of the mRNA with the cap-proximal start codon is not occluded by the rest of the mRNA – either because it binds one or more protein factors or by preventing interactions with other RNA elements. It is certainly an intriguing question, but one we do not have an answer for.

I also assume that the mechanism presented in Figure 5 will only be valid for the first round of translation initiation, since thereafter the mRNA will be translating with most internal structure distorted by the 80S ribosomes. Hence eIF4A-mediated unwinding of global structure should not be required for subsequent rounds of re-initiation.

We agree that the mRNA will likely have a very different structure once multiple ribosomes are translating it in a polysome. One of the limitations of our system (especially when doing single turnover kinetics) is that we can only monitor one round of translation initiation. Although our data might be representative of only the first round of initiation, indeed, the structural elements throughout the message must be resolved for recruitment to take place. At this time we cannot rule out the possibility that the bulk of the work for eIF4A may be to facilitate the first round of translation initiation. We modified the Discussion to state this idea (subsection “A holistic model for eIF4A function in mRNA recruitment”, last paragraph) and note that further experimentation with polysomes would be of great interest in the future.

How do the authors reconcile their data with those of Christopher Fraser's group where substantial initiation events were observed on (CAA)4-5'UTR betaglobin-β-globin mRNA (PMID 28559306, Figure 2) in the absence of eIF4A?

The two types of mRNAs used in the study cited are short CAA-repeat mRNAs, much like the CAA 50-mer in our manuscript, and the Globin-Luc mRNA, encoding luciferase with a globin 5′ UTR. In both Sokabe and Fraser, 2017 as well as our manuscript, the CAA-repeats mRNA could be recruited in the absence of hydrolyzable ATP; however, we also show that the kinetics were faster when hydrolyzable ATP (and eIF4A) were present. In contrast, Sokabe and Fraser were not able to follow kinetics and thus could not discern effects on rates. In their equilibrium studies, Globin-Luc mRNA, harboring structural complexity, required ATP hydrolysis for recruitment, which is similar to what we see with RNAs 7-10 (Figure 3). Taken together, our data are consistent with the model presented by the Fraser lab and we modified our presentation to emphasize this (Figure 5). Lastly, both we (subsection “holistic model for eIF4A function in mRNA recruitment”, first paragraph) and the Fraser lab point out that unwinding of the mRNA is a possible additional function of eIF4A, not mutually exclusive with the idea that the factor modulates the conformation of the PIC.

[Editors' note: the author responses to the re-review follow.]

Essential revisions:1) While many of the experiments are insightful, the ability of the PIC to stimulate the eIF4A-dependent hydrolysis of ATP is less satisfactory, and should perhaps be dropped from the title. Also, because the experiments were performed with yeast eIF4A, this should be indicated in the title, if it is not changed: "Yeast eIF4A is stimulated by the pre-initiation complex and enhances recruitment of mRNAs regardless of structural complexity".

We are grateful for the suggestion and shortened the tittle of the manuscript to "Yeast eIF4A enhances recruitment of mRNAs regardless of their structural complexity."

2) None of the models explain the difference in the affinity of eIF4A to the PIC in the presence of different mRNAs. If eIF4A is recruited through the eIF4F complex to the cap and moves towards the start codon or is required to change the conformation of the 40S, its affinity is defined by the interactions with the 4F complex or the ribosome. It is not clear how the mRNA concentration should enter here and this should be explained.

As the reviewer correctly points out, we cannot provide mechanistic insight into the exact affinities of eIF4A for other components of the translational machinery. The affinity of eIF4A for eIF4G•4E or for the PIC cannot be inferred from the K1/2eIF4A because it is a kinetic constant, not a thermodynamic one. However, the constant gives insight into how the observed rate of a reaction (k_obs_) changes with respect to the concentration of eIF4A. It is possible that mRNA structure inhibits recruitment (i.e., high ∆G for unwinding) and eIF4A lowers that energetic barrier. RNAs with a higher degree of structure require more enzymatic (eIF4A) activity to lower this energetic barrier and achieve the maximal rate of translation initiation, ultimately resulting in a higher K1/2eIF4A. We apologize for the confusion and we now make this point clearly subsection “eIF4A relieves inhibition of recruitment produced by structures throughout the length of mRNAs”, third and last paragraphs. We also added Figure 3—figure supplement 3, illustrating how a structured and an unstructured RNA might affect the K1/2eIF4A and k_max_ values for mRNA recruitment.

Lastly, we appreciate that the previous version of the manuscript did not thoroughly explain the logic for the mRNA concentrations used and the motivation for titrating RNA. We have added an explanation for why we titrated the mRNA (subsection “The ATPase activity of eIF4A•4G•4E is increased by the PIC”, second paragraph) in the ATPase assays. Also, the Materials and methods section (subsection “mRNA Recruitment Assay”, first paragraph) contains the exact assay conditions and refers the reader to previous literature (Walker et al., 2013; Aitken et al., 2016) further detailing the mRNA recruitment system in yeast.

3) Subsection “eIF4A relieves inhibition of recruitment produced by structures throughout the length of mRNAs”, ninth paragraph. The >30-fold effect by a combination of secondary structures in the 5'UTR and the coding region is not obvious, as it looks like 9-11-fold. Furthermore, even an additive effect (if translated in terms of deltaG) should lead to 3x60 (xfold for construct 6 multiplied by xfold for construct 7) = 180-fold acceleration, compared measured 11 or 9 for mRNAs 8 and 9. Something does not add up here.

We understand the concern of the reviewer and would like to take this opportunity to clarify how the magnitudes of the effects were calculated. Figure 3 compares the fold stimulation of the rate of recruitment in the presence of saturating eIF4A versus in the absence of eIF4A, for each construct. That is, the 60-fold effect for construct 7 is calculated by dividing the maximal rate of recruitment of construct 7 in the presence of saturating eIF4A (k_max_) by the rate of recruitment of construct 7 in the absence of eIF4A (KobsnoeIF4A). In contrast, in the original manuscript drew a comparison of the maximal rates of mRNA recruitment (k_max_) among RNAs 4-9. The 30-fold effect is from comparing the maximal rates of recruitment at saturating eIF4A, (Figure 3—figure supplement 1; compare k_max_ for RNA 4 vs. RNAs 8 and 9). Constructs 5 and 6 contain a hairpin in the 5'-UTR and their k_max_ is lower than that for construct 4 by 2.1-fold and 2.7-fold, respectively. Construct 7 contains structure in the coding region but not in the 5'-UTR and its k_max_ is ~2-fold lower as compared to construct 4. When structure is present in the 5'-UTR and the coding region (i.e. constructs 8-9) the k_max_ is decreased by ~30-fold with respect to the k_max_ measured for RNA 4, thus the effects are more than additive. We agree that the previous version of the manuscript did not elaborate on how each comparison was drawn and in the subsection “eIF4A relieves inhibition of recruitment produced by structures throughout the length of mRNAs” we added parenthetical explanations of how each comparison and calculation of effects was derived (e.g., in the eighth paragraph).

4) Introduction – "[…]no apparent sequence specificity[…]" As noted in the recent publication from the Ingolia lab (Iwaski et al., 2016), there does appear to be a preference for a polypurine sequence.

We are grateful to the reviewers for referring us to this work, which we have now included in the fifth paragraph of the Introduction.

Also in the Introduction – "[…]ATP hydrolysis causes a decrease in affinity of eIF4A for RNA[…]" It would seem that eIF4A•ADP does not bind RNA well, however, the slow chemical step of ATP hydrolysis is associated with tighter binding of RNA as evidenced by either nitrocellulose filter binding or UV crosslinking.

We agree with the reviewers that there is evidence for high affinity states in the catalytic cycle of DEAD-box proteins. However, these states are intermediate in the catalysis and ultimately ATP hydrolysis results in a weaker affinity for the RNA in the ADP-bound state. We added this important point raised by the reviewers as well as the relevant literature references in the fifth paragraph of the Introduction.

5) Concerning the in vivo rate of initiation – it is likely that all in vitro studies that examine only the binding of a single ribosome to a mRNA will likely be an order of magnitude slower than the rate measured in vivo as the in vivo rate could reflect that of polysomal re-initiation, which is much more efficient (that would be a consequence of an intramolecular effect).

Thank you for the insight. We weakened our statement in comparing the rate of eIF4A catalysis with rates of initiation in vivo (Introduction, sixth paragraph).

6) The stimulation of ATP hydrolysis is about 2-fold by the 43S/PIC (Figure 2). This is not very dramatic given the number of components involved, the presence of RNA in the 40S subunit and the presence of components known to enhance eiF4A function (eIF4G/4E and eIF4B). In fact, the limited stimulation seen with added eIF4G/4E may readily reflect that it is present at only 10% of the concentration of eIF4A. Perhaps the most unusual feature of the data in Figure 2, which is the apparent inhibition by either eIF4B or eIF5. Secondly, as noted in panel B of Figure 2, roughly 75% of the ATPase activity of the combination of the 40S/PIC + eIF4A occurs in the absence of added RNA.

The k_cat_ for ATP hydrolysis by eIF4A in the presence of eIF4G•4E is increased 3.4-fold by the PIC, whereas the k_cat_ in the absence of eIF4G•4E is increased 6-fold by the PIC. The latter is bigger than the effect of eIF4G•4E on the k_cat_ of eIF4A in the absence of other components (4-fold). We have now tried to emphasize and clarify these points further by making the comparisons more explicit throughout the manuscript. We believe the confusion arose because the effect of leaving out single components (e.g., 40S subunits) in the presence of eIF4G•4E is only 2-fold. This suggests that the remaining components provide a small amount of stimulation, even in the absence of a complete PIC. When eIF3 and 40S subunits are omitted together, this resulted in nearly a 3-fold decrease in rate, almost to the full 3.4-fold effect cited above. We now make these points more clearly in the third and fifth paragraphs of the subsection “The ATPase activity of eIF4A•4G•4E is increased by the PIC”. In contrast, in the absence of eIF4G•4E, when any single core component of the PIC is omitted (e.g., 40S, eIF2, or eIF3), k_cat_ is reduced by nearly 6-fold, to the level observed with eIF4A on its own, as described in the last paragraph of the subsection “The presence of the 43S PIC and eIF3 increases eIF4A ATPase activity in the absence of eIF4G•4E”. We now also emphasize in the Discussion that the 6-fold stimulation of eIF4A’s ATPase activity by the PIC in the absence of eIF4G•4E is slightly larger than the effect of eIF4G•4E on eIF4A in isolation (Discussion, first paragraph). We apologize that we were not clearer in the previous version of the manuscript.

Removing eIF4B or eIF5 from the reaction did not cause any inhibition, but rather had no effect on the k_cat_ compared to when all of the components were present (k_cat_ ~ 8 min^-1^) (Figure 2, compare "Complete PIC" to "-4B" and to "-5").

The reviewers are correct that it would be interesting to determine if the effects seen here are increased when eIF4G•4E is saturating and stoichiometric with respect to eIF4A. However, we are not able to achieve this scenario for technical reasons, including the fact that eIF4G•4E becomes insoluble at high concentrations. In addition, it should be noted that the concentration of eIF4A in vivo in yeast is ~10-20-fold higher than that of eIF4G (von der Haar and McCarthy (2002) Mol. Micro. 46: 531-544), indicating that our concentration regime is physiologically reasonable.

7) Figure 2—figure supplement 1 – previous studies (see also Lorsch and Herschlag) have indicated an eIF4A Km value for ATP in the 200-400 nM range. What is so different to cause a change such as that seen in panel D?

The previously reported K_m_s for ATP for yeast and mammalian eIF4A are in the 200-500 µM (not nM) range (see, for example, Lorsch and Herschlag, (1998) Biochemistry 37: 2180-2193). The reviewers are correct that the K_m_s we report here for eIF4A and eIF4F are 2-5-fold higher than previously reported values for the yeast and mammalian factors. We now make this point and suggest that the discrepancies could be due to differences in buffer conditions or the RNAs used to activate the ATPases (e.g., poly(U) or poly(A) vs. structured and natural mRNAs here) in the last paragraph of the subsection “Km values suggest distinct mechanisms of eIF4A activation by the PIC components and eIF4G•4E”. We also make clear that there are some known differences in the behavior of yeast and mammalian eIF4A and eIF4F that could lead to different kinetic constants. We thank the reviewers for suggesting we address these discrepancies.

8) Discussion – an extensive discussion is centered around a limited stimulation of the eIF4A-dependent ATP hydrolysis. This should be considerably condensed. A clear contrast of yeast vs. mammalian systems should be highlighted.

We appreciate the input and did our best to condense the Discussion by taking out as much redundancy as possible while succinctly drawing comparisons, where appropriate, between the yeast and the mammalian systems. We hope the Reviewers will agree that – given the complexity of the system – taking out any more text will cause the results to lose context within the large body of previous work in the field. In its shortened form, this section of the Discussion is now only one double-spaced page.